# Climate's influence on topography encoded in stream network topology and geometry

Minhui Li [1,2,3,4,9], Hansjörg Seybold [5,6,9], Xudong Fu [1,3] ✉, Baosheng Wu [1,3], Peter A. Raymond [4] & James W. Kirchner [2,7,8] ✉

Stream networks express how Earth's hydrologic cycle is embedded within its three-dimensional topography. In a top-down view, a stream network's morphology is often described by its topological connectivity and branching geometry. Although these two characteristics are naturally connected, they have mostly been studied independently, leaving their co-evolution poorly understood. Here, we analyze the topology and geometry of 16,322 5th-order real-world stream networks across the contiguous United States, showing how they are shaped by climate and the evolution of Earth's topography. We find that ~73% of these networks show topological self-similarity in their branching patterns and that small tributaries join larger streams at systematically wider angles. Our analysis further reveals that correlations between climate and network topology observed in other studies are mainly mediated through the climate-dependence of networks' geometric and topographic properties, such as their junction angles and channel slope ratios of merging tributaries. These findings demonstrate the co-evolution of network geometry, topography, and topology under the influence of landscape evolution driven by climatic forcing.

Branching networks form striking patterns in many natural systems and are often characterized by power-law scaling relations that suggest underlying universality[1–5]. Stream networks, in particular, form tree-like structures that collect water, sediment, and solutes from the landscape and deliver them to larger water bodies downstream[6,7]. Horton[8] and Strahler[9] introduced a hierarchical ordering scheme (Fig. 1) that has been widely applied to network studies in geoscience and related disciplines[10–14]. Unfortunately, it provides only limited insight into network topology because it does not describe how streams of different orders are interconnected. However, the connectivity between small and large rivers is crucial for flood propagation[11], riverine biodiversity[15], and the understanding of transport processes[16–18]. Headwater streams, for example, are more

dynamically linked to their surrounding hillslopes and groundwater systems[19,20] and therefore serve as hotspots for channel erosion and biogeochemical activity[13,21]. Here, we analyze the relationships between stream network topology and geometry, and explore how climate influences these properties through the networks' embedding in Earth's three-dimensional topography.

To quantify stream network topology (the connectivity between streams of different orders), Tokunaga[22,23] expanded the Horton-Strahler ordering scheme by introducing the concept of side branches (streams of order $\omega$ entering streams of higher order $\omega'$, with order differences $k = \omega' - \omega$, $k > 0$), and bifurcations (streams of the same order merging at a junction, with order difference $k = 0$; see Fig. 1a). By assuming scale invariance for the number of streams of order $\omega$

[1]Department of Hydraulic Engineering, Tsinghua University, Beijing, China. [2]Department of Environmental Systems Science, ETH Zurich, Zurich, Switzerland. [3]Key Laboratory of Hydrosphere Sciences of the Ministry of Water Resources, Tsinghua University, Beijing, China. [4]Yale School of the Environment, Yale University, New Haven, USA. [5]Department of Civil, Environmental and Geomatic Engineering, ETH Zurich, Zurich, Switzerland. [6]Institute for Interdisciplinary Mountain Research, Austrian Academy of Science, Innsbruck, Austria. [7]Swiss Federal Research Institute WSL, Birmensdorf, Switzerland. [8]Department of Earth and Planetary Science, University of California, Berkeley, CA, USA. [9]These authors contributed equally: Minhui Li, Hansjörg Seybold. ✉e-mail: xdfu@tsinghua.edu.cn; kirchner@ethz.ch

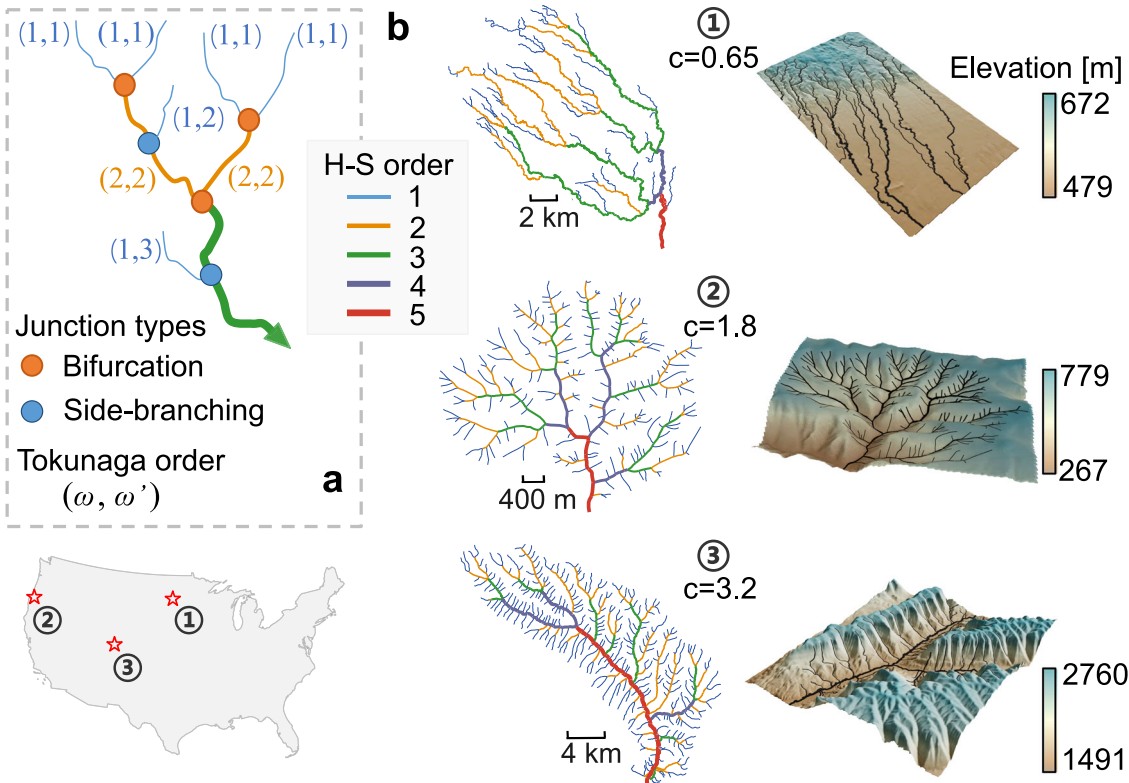

**Fig. 1 | Illustration of the Horton-Strahler and Tokunaga ordering systems, with example river networks. a** The lines represent rivers within a 3rd-order stream network, differentiated by colors corresponding to their Horton-Strahler (H-S) orders. Each pair of Tokunaga orders $(\omega, \omega')$ indicates a stream's H-S order $(\omega)$ and the order of the stream that it meets at its downstream end $(\omega' \geq \omega)$. Junction types are defined by $k$, the (unsigned) difference in H-S orders between the two incoming tributaries: bifurcations occur when $k = 0$, whereas side-branching junctions occur when $k > 0$. **b** Three 5th-order networks with different Tokunaga parameter $c$ values and examples of their topographic embedding. The locations of these three networks are indicated on the outline map.

---

entering streams of order $\omega + k$, a scaling relation can be derived that depends solely on the order differences $k$, rather than on the network's absolute order:

$$T_k = ac^{k-1} \tag{1}$$

Here, $T_k$ represents the average number of side-branches whose order is $k$ lower than the streams they flow into, $a$ is Tokunaga's constant, and $c$ is known as the Tokunaga ratio (see also Methods). Stream networks that satisfy Eq. (1) are called Tokunaga self-similar networks[23]. Tokunaga parameter $c$ measures the relative frequency of stream junctions that have different contrasts in stream order between their tributaries, covering both small- and large-scale features of the network[24] and thus serving as a scale-invariant metric of the network's topology (Supplementary Text S1 and Fig. S1). As illustrated in Fig. 1b, larger values of parameter $c$ imply a greater abundance of low-order channels joining high-order streams, resulting in a more 'feathered' network. While parameter $a$ varies only slightly across different networks, parameter $c$ is sensitive to climatic influences[24]. Yet, estimates of this climatic sensitivity vary with the area thresholds used to extract topographic flow paths[24], and whether such climatic sensitivity persists at the continental scale remains unresolved.

Previous research on stream network topology has mainly focused on determining whether principles such as topological randomness[25] or optimality[6,14,26] can generate realistic-looking network configurations[27–30]. However, these statistical methods have largely ignored the topographic controls that actively shape network topology[7,31]. Three-dimensional topography results from the interaction between tectonic uplift and climate-driven erosion[32–35], through

which landscapes evolve highly dissected, multi-scale drainage networks (Fig. 1b)[7,36]. As a result, the branching stream networks embedded within a landscape reflect the relative dominance of different forcings, each of which leaves a distinct imprint on how channels initiate, grow, and connect across scales[7,34,36]. Laboratory experiments, computational simulations of landscape evolution, and analyses of topographic flow paths from digital elevation data suggest that climate may affect a network's overall topology[24,37–39]. Testing these findings in a natural system requires stream networks mapped at high resolution across a wide range of climatic conditions. Here, we analyze stream networks from the high-resolution National Hydrographic Dataset[40] (NHDPlus-HR), which is extensively ground-checked and provides the best available continental-scale mapped stream networks[41]. This allows us to re-examine classic questions about how much real-world stream networks exhibit self-similar scaling, and how topography and climate influence their topology[24,25,27–30,38,42].

While network topology describes the connectivity of streams of different orders, it does not capture the network's geometry—namely, the length of segments[5] and the angles between them[8,43–52]. A key geometric measure of branched stream networks is the angle between pairs of upstream tributaries, which may indicate different erosion processes in humid and dry climates[44,47,48,51,52], and also influence basins shapes[53,54]. While stream branching angles are only weakly correlated with the average slopes of the two tributaries[47], they are more strongly correlated with the contrast in slopes at the confluence[49]. According to Horton's geometric model[8], branching angles depend on the ratio between the slopes of the shallower stream and the steeper one (the slope ratio SR). The planform structure of stream networks has also been shown to be related to the channel concavity exponent[55], which

relates the downstream change in channel slope to drainage area. Despite these insights, it remains unclear how climatically mediated landscape dissection influences the branching geometry of stream networks at side-branches and bifurcations, as well as network topology.

To better understand how the planform geometry and topological connectivity of stream networks are embedded in three-dimensional landscapes, we first test Tokunaga self-similarity in 16,322 5th-order real-world stream networks across the contiguous United States using the high-resolution National Hydrographic Dataset[40]. We then analyze relationships among Tokunaga parameter $c$, bifurcation and side-branching angles, slope ratios, and climatic aridity. This information allows us to develop a conceptual framework that explains how climate influences network topology through its impact on topography and network geometry.

## Results and discussion

### Test of stream network self-similarity

Because Tokunaga parameter $c$ assumes topological self-similarity[23,24], we limit our analysis to Tokunaga self-similar networks. Of the 16,332 5th-order stream networks from the NHDPlus-HR dataset, 73% (11,946) pass the Tokunaga self-similarity criteria of ref. 24 (see Supplementary Text S2 and Fig. S2). To determine whether the proportion of Tokunaga self-similar networks is affected by external factors like lithology, climate, or topography, we grouped these networks based on the underlying lithology type, mean aridity index ($\overline{AI}$), network-averaged channel slope ($\overline{S}$), and network-averaged slope ratio ($\overline{SR}$; see Methods). Here, slope ratio is defined as the ratio of the gentler to the steeper slope in each pair of upstream tributaries and therefore ranges from 0 to 1 (see Methods). $\overline{SR}$ values closer to 1 indicate smaller contrasts in slopes between pairs of tributaries. The proportion of Tokunaga self-similar networks shows only minor variation across different lithology types and channel slopes (see Methods and Supplementary Fig. S3a, c), and the percentage of Tokunaga self-similar networks in arid climates ($\overline{AI} \leq 0.22$) is only slightly lower (66.6%) than in more humid ones (73.3–75.2%; see Supplementary Fig. S3b). The proportion of Tokunaga self-similar networks decreases slightly, from 77.1% to 62.2%, as slope ratios increase, suggesting that Tokunaga self-similar networks are moderately more prevalent in more dissected landscapes with greater contrasts in channel slopes (smaller slope ratios; Supplementary Fig. S3d).

The 11,946 5th-order Tokunaga self-similar networks have drainage areas with 20th, 50th, and 80th percentiles of 21 km², 90 km², and 297 km², respectively. To explore how varying stream network scales could affect our results, we also repeated our main analysis on the 3454 6th-order stream networks in the contiguous US, resulting in similar conclusions (Supplementary Text S3 and Table S1).

### Spatial patterns of stream network side-branching

Stream network branching patterns result from the evolution of the landscape they are embedded in, and thus can be used to infer climatic and tectonic factors that shape the landscape[7,32,33,36]. To compare regional patterns of Tokunaga parameter $c$ with patterns of climatic aridity, channel slope, and slope ratio, we spatially aggregated these network attributes into 893 equally sized hexagons, each with an area of 10,000 km² (Fig. 2a–d; see Methods). Each hexagon contains, on average, thirteen 5th-order Tokunaga self-similar networks. Less feathered networks (dark colors in Fig. 2a) are more prevalent in arid regions (dark colors in Fig. 2b). Individual hexagons with less feathered networks are found along coastlines and in the glaciated upper Midwest (Fig. 2a), presumably reflecting local geomorphic controls[42]. The binned scatterplot in Fig. 2e shows a slight tendency for Tokunaga parameter $c$ to increase with $\overline{AI}$ (see Methods; higher $\overline{AI}$ values indicate more humid conditions), with the raw data exhibiting a weak but statistically significant rank correlation ($\rho = 0.12$, $p < 0.0001$). Note that all

Spearman rank correlations in this study are calculated based on the raw data, and binning was only used for visualization.

Notable similarities in the spatial patterns of Tokunaga parameter $c$ and slope ratios are observed across the US, with basins having slope ratios closer to 1 (dark color in Fig. 2c) generally corresponding to small values of parameter $c$ (dark color in Fig. 2a). This indicates that basins with smaller contrasts in channel slopes tend to have less feathered networks ($\rho = -0.25$, $p < 0.0001$; Fig. 2f). By contrast, there is only a weak ($\rho = 0.06$), although still statistically significant, spatial correspondence between Tokunaga parameter $c$ (Fig. 2a) and network-averaged channel slope (Fig. 2d, g).

### Interrelations of stream network topology and planform geometry

The planform configuration of stream networks is characterized not only by their topology but also by their branching geometry, which is quantified here by the angles formed at each junction. To explore whether junction angles differ between bifurcations (where tributaries of equal order meet) and side-branches (formed by tributaries of different orders), we split our dataset into its 876,903 bifurcation angles and 1,956,624 side-branching angles. We then examined how these angles vary with AI, mean channel slope, and Tokunaga parameter $c$.

Network-averaged side-branching angles systematically widen with increasing degrees of side-branching as quantified by Tokunaga parameter $c$ ($\rho = 0.25$, $p < 0.0001$; diamonds in Fig. 3a). By contrast, network-averaged bifurcation angles vary only weakly with parameter $c$ ($\rho = 0.01$; circles in Fig. 3a). The average of all junction angles (bifurcations plus side-branches) follows the general trend of the side-branching angles ($\rho = 0.20$, $p < 0.0001$; triangles in Fig. 3a), because they make up over two-thirds of all network junction angles (Fig. 3b). Both bifurcation angles and side-branching angles tend to be wider in humid landscapes than in arid ones[47–51] (Fig. 4a, c), while steep channel slopes lead to narrower angles[47,52] (Fig. 4b, d). Similar patterns are seen when network-averaged differences in Horton-Strahler order are used as an alternative measure of network topology (see Supplementary Text S1 and Fig. S4).

Why are side-branching angles wider than bifurcation angles, and why do they vary more systematically with Tokunaga parameter $c$? As Fig. 3b shows, junctions with larger differences $k$ in Horton-Strahler orders are more common in networks with higher Tokunaga parameter $c$ values. Junctions with larger order differences $k$, in turn, have larger contrasts in the drainage areas and slopes of their tributaries (as reflected in slope ratios farther below 1; Fig. 3c). This directly implies greater contrasts in channel slopes at side-branching junctions ($k > 0$) than at bifurcation junctions ($k = 0$), and greater contrasts in drainage area (and thus channel slope[50]) in networks with greater degrees of side-branching (as measured by higher values of Tokunaga parameter $c$). Pairs of tributaries with larger differences in order tend to have larger differences between their upstream slopes (i.e., smaller slope ratios SR) as a direct consequence of the power-law relationship between drainage area and channel slope[56], combined with Horton's exponential relationship between order and drainage area[8]. Consistent with Horton's steepest-descent geometric model[8], tributaries with larger order differences $k$—and thus larger differences in their drainage areas and slopes—have wider average junction angles (Fig. 3d). This observation leads directly to the expectation that side-branching angles should be wider than bifurcation angles, and should be wider in networks with greater degrees of side-branching (larger values of Tokunaga parameter $c$).

Slope ratios and their corresponding junction angles are topographic expressions of landscape dissection. A valley's high-order main stream will carry a relatively large discharge and therefore adjust toward a relatively low equilibrium channel gradient[34], whereas its side slopes will remain steep unless the streams draining those slopes have sufficient drainage areas (and thus discharges) to incise them[7]. Thus,

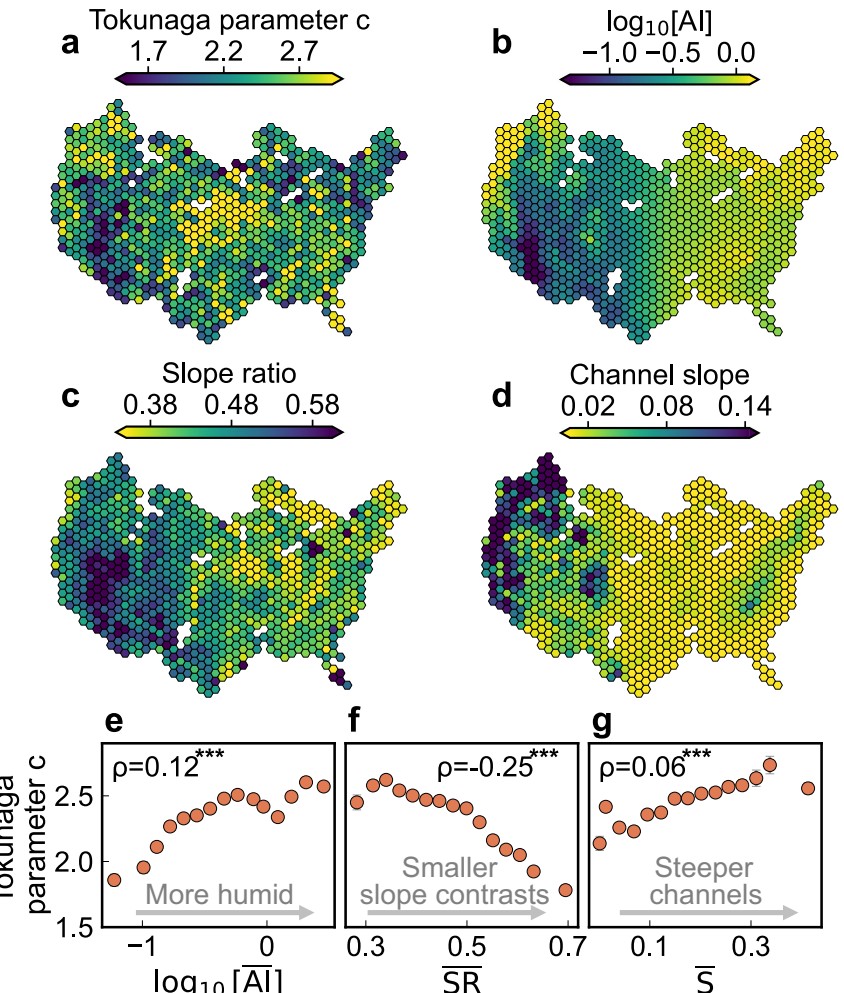

**Fig. 2 | Spatial patterns of Tokunaga parameter $c$, aridity index, slope ratio, and channel slope. a–d** Show maps of the median values of parameter $c$ (higher values indicate more feathered networks), $\log_{10}[AI]$ (higher values indicate greater humidity), slope ratio (higher values indicate smaller slope contrasts between pairs of tributaries), and channel slope within each hexagon (see Methods), respectively. Hexagons with no 5th-order Tokunaga self-similar networks are white. **e–g** Show relationships between binned mean values of parameter $c$ and $\log_{10}$-transformed, network-averaged aridity index ($\overline{AI}$), as well as the network-averaged slope ratio ($\overline{SR}$) and channel slope ($\bar{S}$), respectively, averaged into 15 bins of the candidate explanatory variables. The Spearman rank correlation coefficients $\rho$ are calculated from raw data rather than from the binned data shown in (**e–g**). Triple asterisks (***) indicate significance levels of $p < 0.0001$.

more feathered networks, whose tributary junctions will tend to have larger order differences $k$ and greater contrasts in drainage areas and channel slopes, will also tend to have wider average junction angles. From this perspective, slope ratios and junction angles are not only a direct consequence of local erosion asymmetry, but also a broader result of side-branch formation, which ultimately sets a stream network's topology and geometry.

We further analyzed the empirical cumulative distribution functions of the network-averaged bifurcation angles and side-branching angles for different parameter $c$ classes (Fig. 5). Mean bifurcation angles (circles in Fig. 5a) tend to center around 50° independent of $c$, whereas mean side-branching angles (circles in Fig. 5b) range from 65° to 72° and increase systematically with $c$. As parameter $c$ increases, the curves in Fig. 5b also become steeper, reflecting narrower distributions of network-averaged side-branching angles. The patterns in Fig. 5 imply that processes associated with the creation of low-order side-branches, such as lateral erosion[57] may produce junction geometries that differ greatly from those generated by bifurcation.

### Drivers of stream network topology and geometry
To account for confounding effects that Spearman rank correlations cannot detect, we used partial rank correlation statistics (Fig. 6) to quantify interdependencies between stream network topology (i.e., Tokunaga parameter $c$), side-branching geometry (i.e., network-averaged side-branching angles), and climatic and topographic drivers (i.e., climatic aridity $\overline{AI}$, mean channel slope $\bar{S}$, and mean slope ratios $\overline{SR}$). Analyzing these interdependencies is necessary because climate patterns may be associated with topographic dissection and therefore influence network topology only indirectly.

Prior studies have hinted at possible climatic influences on stream network topology. For example, ref. 24 reported correlations between Tokunaga parameter $c$ and precipitation or storm frequency across 408 stream networks extracted from digital elevation models in the United States. However, these relationships weakened or disappeared when coarser channel initiation thresholds (i.e., drainage area $\geq 0.3$ km$^2$) were used[24], raising the question of whether climatic signatures persist in real-world network topology. Our partial correlation analysis (Fig. 6) indicates that climatic aridity has only a weak direct influence on parameter $c$ ($\rho_{partial} = -0.06$), but stronger relationships with average channel slope ($\rho_{partial} = 0.28$), slope ratio ($\rho_{partial} = -0.32$), and mean side-branching angle ($\rho_{partial} = 0.22$). These variables, in turn, are correlated with parameter $c$. This is consistent with the concept that climatic aridity affects channel incision, which in turn controls landscape dissection[7], thereby setting Tokunaga parameter $c$, channel slopes and

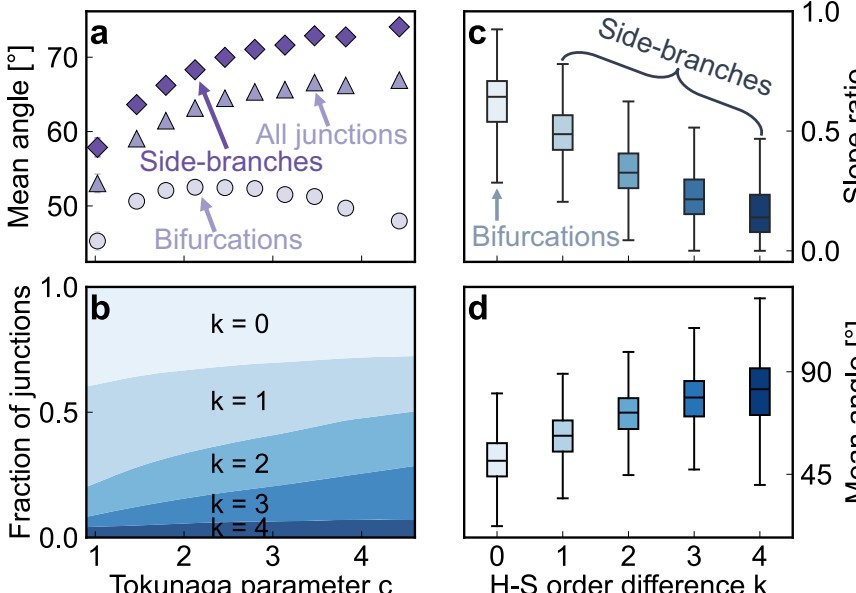

**Fig. 3 | Effects of network topology, as expressed by Tokunaga parameter $c$, on junction angles between tributaries. a** Shows how network-averaged side-branching angles (diamonds), bifurcation angles (circles), and all junction angles (triangles) vary with Tokunaga parameter $c$. **b** Shows how the proportions of junctions with each Horton-Strahler (H-S) order difference $k$ vary with Tokunaga parameter $c$, averaged over the same bins as in (**a**). Boxplots (**c, d**) show how average slope ratios between pairs of tributaries decrease (**c**) and junction angles widen (**d**) with increasing H-S order differences ($k$). **c, d** Boxes represent the interquartile range (IQR) and center lines indicate the median, with whiskers extending to 1.5 times the IQR. Outliers beyond this range are not shown.

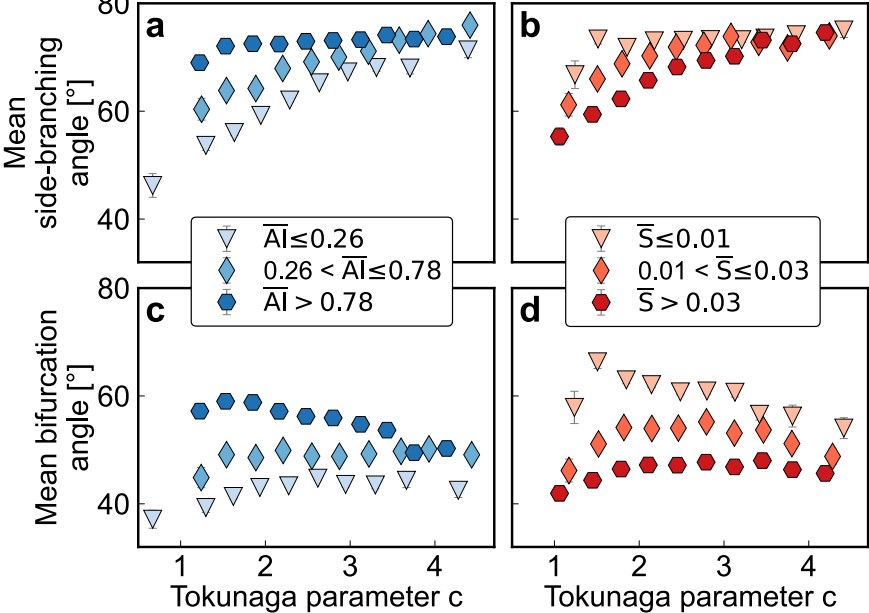

**Fig. 4 | Relation between junction angles and Tokunaga parameter $c$.** Variation in mean side-branching angles (**a, b**) and bifurcation angles (**c, d**) with Tokunaga parameter $c$, across different classes of network-averaged aridity index ($\overline{AI}$) and channel slope ($\overline{S}$). For both aridity index (**a, c**) and channel slope (**b, d**), the first class includes data with values smaller than the 20th percentile, the second class includes the 20th–50th percentiles, and the last class includes values above the 50th percentile.

slope ratios, which ultimately impact side-branching angle. After accounting for climate aridity, slope ratio, and mean side-branching angle, mean channel slope becomes a stronger predictor of Tokunaga parameter $c$ (increasing from an overall correlation of $\rho = 0.06$ to a partial correlation of $\rho_{partial} = 0.19$). This suggests that channel slope exerts a more substantial direct influence on parameter $c$ than climatic aridity does (and that this influence is masked in the correlations reported in ref. 24 and shown in Fig. 2g). These relationships suggest that the observed Spearman correlation between climatic aridity and

Tokunaga parameter $c$ (Fig. 2e) is primarily mediated through climate effects on topography and network geometry (Fig. 6) rather than through a direct impact of climate on network topology.

Stream networks delineate the pathways along which surface water aggregates, and their shapes and structures mirror the landscape's development. Where erosional efficiency is high relative to diffusive smoothing, a branching instability[7] triggers the growth of steep, low-order tributaries with small drainage areas (Fig. 1b). Viewed through this lens, Tokunaga parameter $c$ expresses the prevalence of

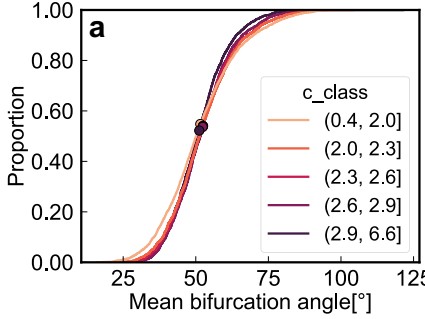
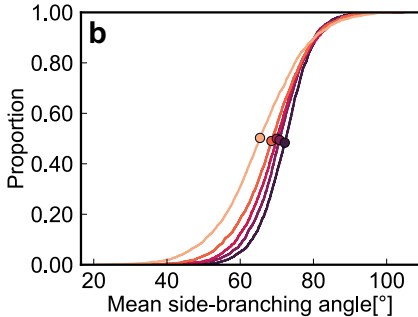

**Fig. 5 | Distributions of network-averaged bifurcation angles and side-branching angles.** Empirical cumulative distribution functions of **a** mean bifurcation angle (between tributaries with the same Horton-Strahler order) and **b** mean side-branching angle (between tributaries with different Horton-Strahler orders), across five ranges of Tokunaga parameter *c*. Solid circles represent the mean angles

for each Tokunaga parameter *c* class. Bifurcation angles are less sensitive to parameter *c* than side-branching angles are. On average, side-branching angles tend to be wider and more narrowly distributed in networks with more low-order side-branches (and thus higher values of *c*).

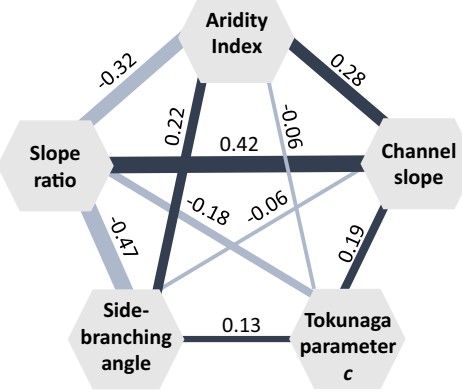

**Fig. 6 | Partial correlations between network-averaged side-branching angles, Tokunaga parameter *c*, and potential drivers.** Negative correlations are indicated in light blue and positive correlations in dark blue, with the line thickness corresponding to correlation strength. Partial rank correlation values are shown beside the lines ($p < 0.0001$ in all cases). Climatic aridity is much more strongly correlated with mean slope ratio, mean channel slope, and mean side-branching angle than with Tokunaga parameter *c*. Parameter *c* is correlated with mean slope ratio, mean channel slope, and mean side-branching angle. Mean slope ratios have a stronger influence on side-branching angles than mean channel slopes do.

this side-branching instability: networks in more strongly dissected terrain exhibit higher *c* values and their junctions exhibit correspondingly stronger contrasts in stream orders, drainage areas (Supplementary Table S2), and channel slopes (Fig. 6). The prevalence of low-order side branches, which tend to run down valley walls perpendicular to the main stream[8,43] (Fig. 1b), influences the statistics of network-averaged junction angles[50] and explains the strong partial correlation between network-averaged slope ratios and side-branching angles ($\rho_{partial} = -0.47$; Fig. 6). However, the effect of average channel slope on side-branching angles is weaker ($\rho_{partial} = -0.06$). Thus, these side-branching angles are more sensitive to how erosion creates contrasts in slope between pairs of tributaries than to the corresponding average channel slopes (Fig. 6). In wetter climates, more effective fluvial incision produces stream junctions with greater contrasts in channel slopes (slope ratios farther below 1; $\rho_{partial} = -0.32$). When topographic effects are factored out by partial rank regression, we find a substantial direct correlation between climatic aridity and side-branching angles ($\rho_{partial} = 0.22$).

Considered together, our observations suggest a conceptual model (Fig. 7) in which climatic aridity may shape both basin topography (i.e., mean channel slope and slope ratios) and network

geometry (i.e., side-branching angles), thereby indirectly influencing network topology (as quantified by Tokunaga parameter *c*). In wetter climates, pairs of tributaries are less likely to share a common slope and orientation, resulting in bigger slope differences and wider junction angles[49]. This observation suggests that the dependence of slope ratios on aridity may partly reflect topological differences in stream networks between arid and humid environments, as noted by ref. 24 and this study (Figs. 2 and 7). Our findings thus illustrate how topography, network topology, and network geometry represent co-evolving fingerprints of climatic influences on drainage basin evolution.

## Methods

### Metric of climate

The aridity index (AI) is the ratio of precipitation to potential evapotranspiration and is widely used as an indicator of climatic wetness (higher AI values indicate more humid climates). We obtain AI values from the Global Aridity Index and Potential Evapotranspiration (ET0) Climate Database, which provides 30-year (1970–2000) climate normals at a resolution of 30 arc-seconds[58]. We expect that the broad spatial patterns of aridity in this 30-year record also reflect long-term spatial patterns, despite shifts in the average climate, barring large-scale reorganization of atmospheric circulation (such that, for example, relatively arid sites have generally remained relatively arid over time[59]).

### Stream networks from NHDPlus-HR

The National Hydrographic Dataset Plus High Resolution (NHDPlus-HR) is a scalable geospatial hydrography framework built from the high-resolution (1:24,000-scale or better) National Hydrography Dataset, the nationally complete Watershed Boundary Dataset, and a 1/3-arc-second (10-m) digital elevation model across the United States[40].

We remove segments labeled "Coastline" from the NHDPlus-HR database, and extract the main flowpath of braided rivers by removing flowlines with divergence flag = 2, yielding networks that have only one downstream segment at each junction. The order $\Omega$ of a network is defined by the order of its most downstream segment, namely the segment that terminates at a lake or ocean, or that merges with another stream of order $\Omega$ or higher. Given the terminal segment, we then identify the upstream network using the 'FromNode' and 'ToNode' attributes in the NHDPlus-HR dataset. This approach allows us to delineate the whole network upstream of any channel with a given Horton-Strahler order. The NHDPlus-HR dataset, however, also includes canals, connectors, and pipelines, which are sometimes relevant to the network's connectivity and therefore cannot be removed easily. Networks where the channel slope is uniformly equal to 0.00001 due to the dataset's slope cutoff are excluded. To mitigate

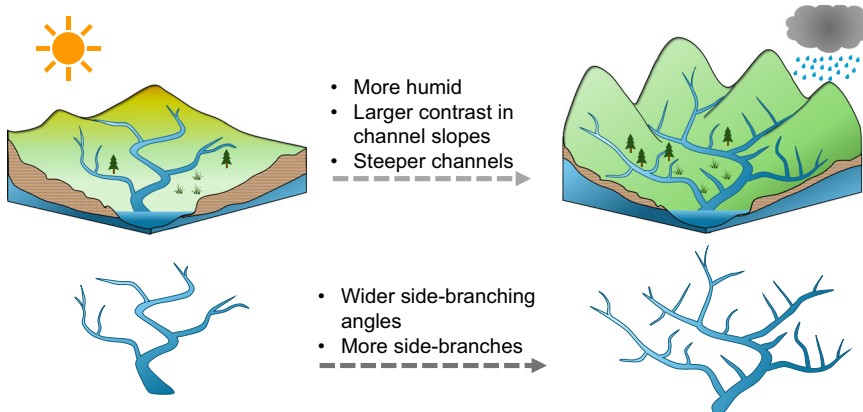

**Fig. 7 | Conceptual diagram illustrating how stream network side-branching structures respond to climatic and topographic conditions.** All else equal, networks with more side-branching and wider side-branching angles are associated with more humid climates, steeper channel gradients, and larger contrasts in channel slopes between pairs of tributaries. Conversely, networks with fewer side-branches and narrower side-branching angles are associated with more arid climates, shallower channel gradients, and smaller contrasts in channel slopes between pairs of tributaries.

the impact of artificial channels, if >10% of a network consists of artificial flowlines (i.e., canals and pipelines with Fcodes of 42800–42817, 42820–42824, 33600, 33601, or 33603), the entire network is discarded. The 10% threshold removes networks with pervasive human disturbance while retaining basins whose artificial channels are limited and spatially localized. Using a threshold of 5% or 20% instead of 10% yields similar results (Supplementary Tables S3 and S4).

The NHDPlus-HR dataset assigns attributes such as Horton-Strahler order and slope values to individual river flowlines. One or more of these flowlines make up each stream segment in the network, that is, each segment connecting a junction or channel head to the next junction downstream (Supplementary Fig. S5). To calculate the slope of each segment, we exclude any flowlines lacking slope data (-0.1% of all the flowlines in 11,946 5th-order networks analyzed in the main text) and then calculate the length-weighted slope for each segment using the remaining flowlines. The slope ratio (SR) at any junction between two stream segments is defined as the ratio of the gentler slope to the steeper slope in each pair of upstream tributaries (which therefore ranges from 0 to 1), so smaller SR values represent larger differences in slopes of the incoming tributaries. SR is then averaged over all junctions in each network to yield the network-averaged slope ratio $\overline{\text{SR}}$, which is used in the main analysis. To obtain the hexagon-averaged values shown in Fig. 2, we extracted AI values for the midpoint of each flowline from the Global Aridity Index and Potential Evapotranspiration (ET0) Climate Database[58]. Given that the Tokunaga parameter $c$ is a network attribute, all midpoints within the same stream network are assigned identical values of $c$. Likewise, all midpoints within the same network share the same network-averaged slope ratio $\overline{\text{SR}}$. Finally, we calculated the median of each variable across all midpoints within each hexagon.

The lithology type of each stream network was determined from the global lithology map dataset from ref. 60. Lithology types were extracted for the midpoint of each flowline, and the lithology type covering more than 50% of the flowlines in each network was identified as the dominant lithology. Networks in which no single lithology type exceeded 50% coverage were classified as having mixed lithology. 301 networks with mixed or missing lithology data or classified as water bodies (-1.8% of all networks) were excluded from the lithology analysis.

### Calculation of Tokunaga parameters
Tokunaga's ordering system extended Horton's ordering by introducing the concept of side-branching using the Horton-Strahler orders of

the two joining tributaries at a junction[22,23]. The Tokunaga side-branching ratio

$$T_{\omega,\,\omega+k} = \frac{N_{\omega,\,\omega+k}}{N_{\omega+k}}, (1 \le k \le \Omega - \omega) \qquad (2)$$

is the number $N_{\omega,\omega+k}$ of streams in a network with order $\omega$ flowing into streams with Horton-Strahler order $\omega + k$, divided by the number of streams $N_{\omega+k}$ with order $\omega + k$. Here, $\Omega$ is the highest order in the network. In self-similar networks, the side-branching ratio $T_{\omega,\omega+k}$ is independent of the order $\omega$ and depends only on the increase in order $k$[23,24], satisfying Eq. (1). In Eq. (1), parameter $a$ represents the mean number of streams of order $\omega$ flowing into streams of order $\omega + 1$, and $c$ denotes how this side-branching ratio grows with increasing contrasts $k$ between the orders of main streams and their tributaries. Tokunaga parameter $c$ thus reflects the degree of lower-order side-branching, with higher $c$ values indicating larger average differences between the orders of side branches and the main streams that they flow into.

To estimate Tokunaga parameter $c$, we follow the method outlined by ref. 24. First, we calculate the side-branching ratio $T_k$ as

$$T_k = \frac{1}{n_k} \sum_{i=1}^{\Omega-k} N_{i,\,i+k} \qquad (3)$$

where $n_k = \sum_{i=k+1}^{\Omega} N_i$ is the number of streams of order larger than $k$. In some cases, $T_k$ can be zero because the network does not contain two tributaries whose difference in Horton-Strahler order is $k$. In these cases, the Tokunaga coefficients cannot be reliably determined, so we discard the entire network.

We then estimate $c$ from linear regression of the log transform of Eq. (1),

$$\log_{10} T_k = (k-1)\log_{10} c + \log_{10} a \qquad (4)$$

using weighted least squares with weights $w_k$ based on the number of branches[24],

$$w_k = \sqrt{\sum_{i=1}^{\Omega-k} \sum_{l=1}^{N_{i+k}} 1} \qquad (5)$$

The weighted least squares method yields a smaller variance for the Tokunaga parameters than that obtained with unweighted least squares[24]. As suggested by ref. 24, statistics for the highest-order rivers

of each network are excluded in the Tokunaga parameter calculation to minimize the influence of finite-size effects.

## Stream network branching geometry

We characterize network geometry using the branching angles between pairs of incoming tributaries, averaged over all junctions. To calculate the branching angles, we use a conformal projection (Lambert conformal cone 102004) to map stream networks from NHDPlus-HR. Adopting the approach described in ref. 47, we converted all river segments to a sequence of discretization points. Then we fitted a straight line through the points of each river segment using orthogonal regression. This method allows us to characterize the average orientations of the tributary valleys and the angles between them[47]. Our analysis encompasses a total of 2,833,527 junctions in 11,946 5th-order Tokunaga self-similar stream networks, and 2,209,988 junctions in 2417 6th-order Tokunaga self-similar networks.

## Scatterplot binning

In our binned scatterplots (Figs. 2e–g, 3a, and 4a–d), the first and last bins comprise x-axis values smaller than the 1st percentile and larger than the 99th percentile, respectively. The remaining bins are equally spaced between these percentiles. The error bars indicate the standard error of the mean for each bin, where these are larger than the plotting symbols. Note that this binning procedure retains most of the variability along the horizontal axes but averages out much of the variability along the vertical axes.

## Reporting summary

Further information on research design is available in the Nature Portfolio Reporting Summary linked to this article.

## Data availability

The National Hydrographic Dataset Plus High Resolution (NHDPlus-HR) is available from https://www.usgs.gov/national-hydrography/nhdplus-high-resolution, and the Aridity Index dataset is available from Trabucco & Zomer (2019)[58] (https://doi.org/10.6084/m9.figshare.7504448.v3). The global lithology map dataset is available from Hartmann & Moosdorf (2012)[60](https://doi.org/10.1594/PANGAEA.788537). The datasets used to produce our results are available at https://doi.org/10.5281/zenodo.18627184.

## Code availability

The code to reproduce the main results of this study is available at https://doi.org/10.5281/zenodo.18650981.

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

## Acknowledgements

The authors thank S. Mudd and three anonymous reviewers for feedback and insights. X.F. acknowledges support from the National Natural Science Foundation of China (No.U2340223). H.S. acknowledges support from the EU Horizon CryoSCOPE project supported by the State Secretariat for Education, Research and Innovation (grant number: 101184736). P.R. acknowledges support from the NASA Surface Water and Ocean Topography program (grant 80NSSC24K1654).

## Author contributions

M.L. and H.S. conceived the study. M.L. led the methodology development, investigation, visualization, writing of the original draft, and editing of the manuscript. H.S. contributed to methodology development, morphological interpretations, and manuscript editing. X.F. and B.W. supported the study. X.F. and P.R. contributed to the discussion and editing process of the final manuscript. J.K. contributed to the conceptualization, interpretation, discussion, review, and editing of the work. X.F. and J.K. supervised the work.

## Funding

## Competing interests

The authors declare no competing interests.
