## [Transparent Peer Review file · Nature Communications]

Climate's influence on topography encoded in stream network topology and geometry

Corresponding Author: Professor James Kirchner

Version 0:

Reviewer comments:

Reviewer #1

(Remarks to the Author)
23 September 2025

Review of "Climate's influence on topography encoded in stream network topology and geometry"

In the submitted manuscript, the authors use a recent hi-resolution dataset of stream networks across the USA to identify how climate affects stream network topology and network geometry. Specifically, they identify that increases in the branching angle of river networks is accommodated through increases in the branching angles of "side branches", not bifurcation branches. The results presented here show that, in self-similar networks, the angle of side branches records variation in both the slope ratio of streams at confluence, and the aridity index of a landscape.

Overall, I believe this manuscript represents a significant contribution to the scientific community and its potential impact to a diverse cross-disciplinary audience justifies its publication in a high impact journal such as Nature Communications. I do have some comments regarding the methodology, clarity of text, and discussion points which I outline below.

Major Comment: Adding mechanistic explanations of the results

The largest shortcoming of the manuscript is its lack of mechanistic explanation for the observed results and correlations. Numerous times the authors stated (or implied) that one measured quantity drove another based on a correlation, without providing a reasonable mechanism to explain it (i.e., there is correlation, but no causation). For example, L198-201 argues slope ratios may influence side branching and thus topological connectivity, as well as branching geometry. While I agree with the authors' findings that these network characteristics and metrics are correlated, a mechanistic explanation significantly strengthens the argument that one drives another. I will not go through every example of this in the manuscript, but other instances of this can be seen in L271-274, L309-312 and L315-317. Addressing this will provide support for the authors' argument, starting in L76, that physical processes are considered in this contribution.

Minor Comments

Manuscript structure:

The following is a suggestion which the authors can ignore if they choose: I think this manuscript would be easier to read (and thus more impactful) without using or minimizing the use of self-similarity and Takunaga's parameter c . After reading the manuscript, I believe the only benefit of including Takunaga's parameter c is to show that increases in the average side branching angle are a result of actual branching angles increasing and not a result of basins with inordinate amounts of large delta HS confluences (i.e., there are two ways to increase the average side branching angle in a basin: have more large delta HS confluences relative to small delta HS confluences, or increase the branching angle of confluences). However, this same point, I think, could also be made in a different way which I believe will be more intuitive for most readers and will not take away the scientific rigor of the approach. Instead of using Takunaga's parameter c as the x axis in Figure 3, I think these results could be broken up by delta HS values or potentially normalized by the average delta HS within the basin. This approach would clearly show the reader that increase in the average side-branching angles correlated with Aridity Index or average slope are a result of actual angles of the confluences increasing. If the authors feel that Takunaga's parameter c is still better, perhaps an alternative is to show both? One of the metrics could be shown in the supplement if space a concern.

Methods:

- Why was 10% of the network consisting of artificial channels chosen as a threshold cutoff? While a sensitivity analysis would be helpful to show the influence of artificial channels on the results, I recognize this requires significant effort. If a sensitivity analysis is not performed in a future submission, it would be helpful to provide justification for the 10% cutoff threshold.
- More information about the lack of slope information connected with certain stream segments would be helpful for the reader. How many stream segments lack slope information? Is there a spatial trend to the lack of slope information? Similar to the previous comments, how does this impact the results?

Line and Figure Comments

- Line 52-56: It appears the variable k , may represent two things. Text on line 52 implies k may be the order of the tributary, while Lines 58-59 clearly define k as the difference in Strahler stream orders. Can you please clarify this text?
- Line 84-86: How are channel heads identified if a threshold drainage area is not used?
- Line 87-90: I would suggest either softening this language or describing the exact question that is answered here in more detail. Many workers have done substantial work on this topic, and many questions are left. Please give citations for the long-standing questions as well.
- Line 96: Please ensure at least one of these citations is appropriate support for the topology and geometry argument in this sentence. From my recollection the cited references mainly discuss the morphologic characteristics, but I could be wrong.
- Line 121: More information would be helpful to let the reader know why a two-step approach is necessary here. It is unclear how a river can be self-similar but not meet Takunaga's scaling criteria.
- Line 131: Although the slope ratio is well described in the methods, SR is a major component of this story and, for the reader to understand the results described here, a better main text explanation of SR would be helpful.
- Line 144-146: If c is kept in the manuscript, I suggest moving this description of c to earlier in the text. The variable is referred to frequently and a description such as this helps readers to conceptualize the variable.
- Line 152: Can you provide a citation?
- Line 180-182: I suggest moving this description of side branching earlier. It will help the reader understand the arguments presented before this point.
- Line 320: I do not believe the analysis provided in this manuscript provides support for this statement. If the authors would like to argue that side-branching angles are not controlled by tectonics, a more robust analysis of this is required. I would suggest removing the text "overall rates of uplift and" and change steepness to slope. The word steepness suggests the normalized metric, channels steepness, was used in this analysis.
- Figure 3: Having the same symbols in panel b represent something different than they do in panels c-f is confusing for the reader. Please change.

(Remarks on code availability)

I only very briefly looked at the code. The readme file seemed to be empty and did not have instructions. I saw code was present in the other file, but I did not inspect it.

Reviewer #2

(Remarks to the Author)

(Remarks on code availability)

Code appears to produce plots shown in manuscript but does not contain the more detailed analyses of channel networks.

Reviewer #3

(Remarks to the Author)

REVIEW – NCOMMS-25-68072

This manuscript sets out a compelling data-driven argument for how climatic signature is expressed in river network topology – not directly, but through mediating conditions of basin topography and network geometry.

The analysis draws upon a collectively extensive body of work that has explored each of these contributing components – topography, network geometry, and network topology – respectively in relation to climatic forcing without quite disentangling the nature of their inter-relationship. This contribution gets closer to that aim, leveraging both a high-resolution dataset of channel networks for the contiguous USA, and a clever application of partial rank correlation statistics to unpack subtle but significant indirect links through which climatic forcing is imbued in network topology.

My remarks on this manuscript are relatively minor, but pertain to the narrative. I hope my comments might help the authors clarify their message that much more, for their readers' benefit.

The first sentence of the final paragraph (L327–329) presents the clearest summary framing (problem and findings) in the manuscript – so much so that I think it should get worked into the introduction. To me, the introductory paragraphs dive too quickly into how topology and topography are quantified, and there are some intermediate punchlines (e.g., L88–89) that are confusing. (I understand that the NHDPlus-HR dataset allows previously inaccessible insight into self-similarity in real river networks, and therefore robust application of Tokunaga ordering. The authors should absolutely point this out – but as written it diverts the reader from the larger aim; it's an important finding, but also a means to an end.)

I wonder if the authors might shift the paragraph at L106 up to be the second paragraph (inset at L49). The paragraph on Tokunaga ordering (L49) could introduce the Results section at L117. For continuity, I would also put the paragraph at L141 above the section break, to be the final paragraph of the first Results section – that is, insert after L139. Similarly, the paragraph on topography (L91) sets up the Results section at L147, and could serve as the opening paragraph there.

I really enjoyed the synthesis section from L252 onward, and I think it's one of the strengths of this manuscript. In that vein, I encourage the authors, when they re-read this work with fresh eyes, to do their best to make each paragraph a ratchet that moves the reader inexorably forward. (Put another way, be on the lookout for recursions in the text.)

I look forward to seeing this work in print – congratulations to the authors on a fine analysis.

(Remarks on code availability)

N/A

Reviewer #4

(Remarks to the Author)

This paper looks at relationships between networks topology, channel gradients, lithology and climate. The paper shows that gradients are more highly correlated with a metric called the Tokunaga ratio than they are with climate. I have a number of questions related to the analysis but overall I think this is a very interesting paper and shows something new. One of the most interesting aspects of this work is that the Tokunaga ratio is correlated with the side branching angle and not the bifurcation angle, suggesting that the process of adding smaller links leads to different channel geometry than bifurcation. To me this has interesting implications for network growth, has never been shown before, and somehow the authors have not deemed it important enough to mention in the abstract. I would change that.

I think the choice of using 5th order basins vs 6th or 7th order could be clearer. On the other end of the size spectrum, I think a comment about how many 5th order basins cross big physiographic divides might be in order (that is, does the basin size lead to many basins crossing from steep mountains into gentle basins). The reason for a comment on this is the somewhat strange pattern in the slope map (see comment below).

Overall I think this is a very interesting contribution, raises some new questions about network topology, and I would characterise most of my comments as cosmetic.

Lined comments

Line 64: You later say that you cannot reproduce this result (or at least you think climate is of secondary importance). The way this is written seems (to me at least) that you are taking this result as a given, when in fact there is only one paper that says this and that your work does not support it.

Figure 1: A this point in the paper ω is not defined. It should be somewhere (perhaps above equation 1). Also the labelling is inconsistent (are you labelling nodes or edges, and if one or the other why aren't all of them labelled?).

Line 81-90: The novelty of this paper (and I think it is quite novel) is the analysis. It is not the underlying data. These lines have many problems and are not necessary for the manuscript.

The report cited here has been superseded by a 2025 report (Moore et al., 2025). It does not really explain how channel heads are found in the NHDPlus HR dataset. It does explain the many corrections needed to be made to the underlying NHD data because once the lidar was collected it emerged that many of the channels mapped on USGS quads were not actually in the right place (see page 60 of the report). So the NHDPlus HR is already corrected by topographic data. The authors then say that other datasets use a threshold drainage area, which leads to inconsistent channel networks. But as the report from Terziotti and Archuleta (2020) makes clear, the NHD is also inconsistent (from their page 16):

“The original NHD was digitized from individual 7.5-minute quadrangle map sheets that were compiled at different times, by many individuals, using varied sources; therefore, some areas of the country have hydrography that is represented at different densities. These discrepancies are due to differing source material or standards and procedures and are not due to differences in geomorphology or hydrologic conditions.”

This report goes on to explain how these differences in channel heads would be fixed, and advocates for a variety of methods for repositioning channel heads. They will insert “additional features” (including new channel heads)

- if there is clear evidence of the feature in the elevation data source,
- if there is clear evidence of the feature using an appropriate ancillary data source,
- if a method has given good results for delineation of stream channels or other features, and it is quality assured using the elevation data and other high-quality ancillary datasets”

So the overall impression is that the NHDPlus HR is hand curated in some way that is not reproducible and is just as susceptible to bias as an algorithmically-derived channel network. The user guide of Moore et al (2025) doesn't explain how this is done so we are left to guess (unlike networks that use a threshold drainage network and the threshold drainage area is reported). I have a strong suspicion the results using a threshold extraction or a more advanced channel head finding algorithm on the NDEP lidar data would result in outcomes indistinguishable from those reported here, NHDPlus HR is as good a network as any (and if you think hand curation is better, it might be better), and I don't think there needs to be any modification to the analysis. But this paragraph makes misleading statements and needs to go. Just say “The channel network used was the NHDPlus HR dataset”.

Moore, R.B., McKay, L.D., Rea, A.H., Bondelid, T.R., Price, C.V., Dewald, T.G., Hayes, L., 2025. User's guide for the National Hydrography Dataset Plus High Resolution (NHDPlus HR) (No. 2025–5031), Scientific Investigations Report. U.S. Geological Survey. <https://doi.org/10.3133/sir20255031>

Terziotti, S., Archuleta, C.-A., 2020. Elevation-derived hydrography acquisition specifications (No. 11-B11), Techniques and Methods. U.S. Geological Survey. <https://doi.org/10.3133/tm11B11>

Line 108: Why 5th order? In 5th order channels, when you do the regression to calculate c you are only fitting 4 points. Do your results change with 6th order basins, for example?

Line 120-122: The explanation of the ANOVA test is missing from the methods. I don't think it will change the result, but why was a non-parametric test not chosen for this step? Zanardo et al have an entire paragraph about the drawbacks of the ANOVA (paragraph 31 in their paper) and I never understood why they didn't use a non-parametric test.

Line 130-132: Climate varies over a longer wavelength (loosely defined) than lithology. That is, in a 5th order channel network you are unlikely to have a large variation in the aridity index, but you could have several rock types with very different properties. The rock hardness is quite strongly correlated to the various slope metrics. The lithology is assigned of 50% of the underlying rocks are the same in a catchment, but harder lithologies in the headwaters and softer lithologies downstream will have quite a different effect, I imagine, on topology than vice versa. Some comment on this would be welcome.

Line 130-131 and Line 371-373: I find it counterintuitive to select a slope ratio, which is a metric specifically designed to test differences in gradients in streams of different order, in such a way that a larger value represents a smaller variation in channel gradients. Surely this metric should be chosen so a higher slope ratio represents a larger variation in channel gradients.

Lines 143-149: Can you please report the distribution of sizes of the basins. How many are in each 10,000km² hex?

Line 152: You mention here clustering of more feathered networks in specific geomorphic settings, and then say this “presumably reflects geological controls”. If you think it is glaciers causing the changing topology then you are implying it is *not* a geological control.

Figure 2f: I presume, because you are using NHDPlus, the more arid catchments have a lower drainage density. The slope ratio between two parts of a drainage network depends on the ratio of drainage area raised to the power of the concavity index. We know there is a climate control on the concavity index. Is the signal in panel f controlled by this or by the difference in area ratios because of the changing drainage densities?

Figure 2d: I am puzzled by the slope map. The southern tip of the Appalachians appears to have the similar gradients to the most tectonically active parts of the west coast. This makes me a bit concerned that the size of the hex squares is introducing some bias (i.e., if the hex square is centred on a hilly region, vs a very mountainous region that have half the square on a low relief area (easy to do in, say, southern California), you might get a result not really reflective of the true gradients of the channels. This fear would be mitigated by some analysis of the basin sizes.

Line 186-187: This sentence implies that the 72 degree result is general, and that it has been observed in seepage networks, when in fact the 72 degree result is based specifically on a model including seepage. Do you have a reference for the 72 degrees resulting from a process other than seepage?

Line 195: The relationship between slope ratios and delta HS is controlled by the area ratio, so I don't see what the work “also” is doing in this sentence.

Line 284: Add “the” at the beginning of this sentence.

Line 199: Can you explain how the degree of landscape dissection is reflected in the slope ratio? Are you trying to make the point that a more feathered network has greater area ratios at tributaries?

Line 227: Again, the 72 degrees is the theoretical prediction only for stream sapping.

Line 239-240: “tend to reflect distinct erosion mechanisms”. Reference 53 proposed a very specific erosion model and then went to one of the very few places on Earth where the model could be applied to real topography (Apalachicola is as close to a pile of noncohesive sand as it is possible to get in a natural landscape). And since many of the authors of this paper were also authors of reference 47, they should know that they did not compile erosion mechanisms, but rather speculated on this based on climate regimes. Soften this statement.

Line 264: Replace the word “these” with “their” to make it clear this is a result from the zanardo et al paper.

S Mudd

(Remarks on code availability)

Version 1:

Reviewer comments:

Reviewer #1

(Remarks to the Author)
8 February 2026

Second Review of "Climate's influence on topography encoded in stream network topology and geometry"

I have now read through the revised version of this manuscript, including the supplement and response to reviewer comments. I believe the revised manuscript represents an improvement upon the original submission, and I am satisfied with the author responses to my previous comments (and corresponding changes to the text). Overall, I think this is an interesting contribution that can be published as is. At this point, I only have a few very minor comments that I have listed below, and which the authors can choose to take or leave as they see fit. I'd like to thank the authors for their thorough response to my previous comments, and for sharing their exciting science.

Minor comments:

L214-223: Parts of this explanation were a bit hard for me to follow and I had to take my time with it. Here's how I think about it (which I think may be a bit more intuitive than the text in the manuscript). Side-branches need to have higher contrasts in slopes (lower SR) than bifurcations due to the scaling of slope with drainage area. This dictates that side-branching angles should generally be wider than bifurcation angles, since a wider angle minimizes the portion of the main valley along which the side branch flows. Consider a thought experiment where a side branch comes in with a tiny (let's say 1 degree) junction angle with the main stem. In this case, both the side branch and the main stem must be flowing along the central valley axis. Thus, the topography would need to be really strange for the side branch to have a steeper slope than the main stem. But, if the side branch doesn't have a steep slope, there's likely not enough discharge (for that low drainage area of the side branch) to move the imposed sediment load. In contrast, if the side branch comes in at a 90-degree junction angle, there is no overlap between the side branch and the main valley axis, allowing a low SR. Aspects of this come up in L238-247, but even then, I don't feel like those lines fully explain this concept.

Fig. 4: In Fig. 4, red colors correspond to high AI, which is a humid climate. I (and I think most other people?) tend to think of red as warm/arid. It might help to flip the color scheme for 4a and 4c and make red the low AI (arid) color and make the high AI (humid) bin a cooler color (blue?).

While I found the authors analyses to be robust, one thing that struck me, especially with Figure 7, is that drainage density isn't discussed in the manuscript. Doesn't the increase in Tokunaga parameter c with AI (higher c in more humid climate) imply an increase in drainage density (i.e., higher drainage density in more humid climate)? If so, this seems at odds with studies showing drainage density decreases with increases in mean annual precipitation (e.g., Chadwick et al, 2013). This could be worth a few lines of discussion in the manuscript.

Chadwick, Oliver A., Josh J. Roering, Arjun M. Heimsath, Shaun R. Levick, Gregory P. Asner, and Lesego Khomo. 2013. "Shaping Post-Orogenic Landscapes by Climate and Chemical Weathering." *Geology* 41 (11): 1171–74.

Reviewer #3

(Remarks to the Author)

Thanks to the authors for their considered and substantive revisions to this submission. I have no further comments, and am happy to see this work advance.

Reviewer #4

(Remarks to the Author)

I am satisfied with the authors' responses to my previous comments. However they have added some text in response to another reviewer that I don't agree with (and mis-represent the findings from one of my papers). However this is easily corrected and constitute minor revisions.

Line 62: Insert "The" before Tokunaga. Same on line 126.

Line 133-134: Say which channels are being compared the SR here (not just in methods).

Lines 159-162: Consider making a comparison to reference 24 here.

Line 220-231: Reference 50 has a long discussion of these effects so I am surprised that paper is not cited here.

Line 252: Reference 50 showed that the junction angles were a function of the area ratio, which could have a large range in "side-branching" channels. Lumping all side branching channels seems like a good way to hide interesting behaviour.

Line 254-256: I don't agree with these statements, and it mis-cites our paper. Lateral erosion does not produce low-order

side branches. In the case of very large area ratios, where the larger channel meanders, there is a greater probability that a side tributary will intersect with a meander apex. Here is the text from our paper (reference 50):

The complex hydrodynamics of river confluences and associated sedimentation and erosion (e.g., ref. (69) and references therein) may plausibly promote the lateral migration of network junctions in the direction of the smaller tributary, causing the larger river to form a bend, with the tributary joining at the outside apex. This explanation seems unlikely for two reasons. First, our junction angle measurements reflect the network-scale structure rather than the functional scale of confluence hydrodynamics. Second, it seems inconceivable that the paths of very large channels could be meaningfully impacted by channels orders of magnitude smaller.

What appears as the “deflection” of large rivers by minor tributaries may alternatively be interpreted as minor tributaries preferentially joining large rivers at the outside apex of large bends. In other words, the location of the junction may be influenced by the location of the bend, rather than vice-versa. This interpretation is supported by the observation that developing tributaries nucleate at the outside apex of bends when tidal channel networks form (70).

Response to reviewers' comments

Reviewer #1

In the submitted manuscript, the authors use a recent hi-resolution dataset of stream networks across the USA to identify how climate affects stream network topology and network geometry. Specifically, they identify that increases in the branching angle of river networks is accommodated through increases in the branching angles of “side branches”, not bifurcation branches. The results presented here show that, in self-similar networks, the angle of side branches records variation in both the slope ratio of streams at confluence, and the aridity index of a landscape.

Overall, I believe this manuscript represents a significant contribution to the scientific community and its potential impact to a diverse cross-disciplinary audience justifies its publication in a high impact journal such as Nature Communications. I do have some comments regarding the methodology, clarity of text, and discussion points which I outline below.

Response: We sincerely thank the reviewer for the positive assessment of our study and for recognizing its potential to advance understanding of how climate influences stream network topology and geometry. We have carefully considered all the reviewer’s constructive comments and have revised the manuscript accordingly. Detailed responses to each point are provided below.

Major Comment: Adding mechanistic explanations of the results

The largest shortcoming of the manuscript is its lack of mechanistic explanation for the observed results and correlations. Numerous times the authors stated (or implied) that one measured quantity drove another based on a correlation, without providing a reasonable mechanism to explain it (i.e., there is correlation, but no causation). For example, L198-201 argues slope ratios may influence side branching and thus topological connectivity, as well as branching geometry. While I agree with the authors' findings that these network characteristics and metrics are correlated, a mechanistic explanation significantly strengthens the argument that one drives another. I will not go through every example of this in the manuscript, but other instances of this can be seen in L271-274, L309-312 and L315-317. Addressing this will provide support for the authors' argument, starting in L76, that physical processes are considered in this contribution.

Response: We thank the reviewer for this comment and we have expanded our discussion to describe the physical processes behind the correlations between topographic metrics (such as slope ratios), geometric measures (such as junction angles), and measures of topology, namely Tokunaga parameter c . As explained in the “*Interrelations of stream network topology and planform geometry*” and “*Drivers of stream network topology and geometry*” sections of the revised manuscript, the shape of the river network results from the evolution of the landscape, which determines how channels become integrated within the surrounding three-dimensional topography. We now explain how the topology of the network is related to the drainage areas and slopes of tributaries that meet at bifurcations and side-branches, and show how this affects the distributions of bifurcation and side-branching angles.

Specifically, we have made the following changes:

1. **See Lines 214-247 in the revised manuscript**

Lines 198-201 of the original manuscript said: “*These results suggest that the degree of landscape dissection, as reflected by slope ratio, may influence the formation of side branches and consequently shape their topological connectivity and average branching geometry within river networks.*” To make the connection among network topology, branching geometry and slope ratio more explicit, we have added new panels to the revised Fig.3 (see Fig. 3 of the revised manuscript or below) and expanded our discussion of the underlying physical mechanisms (see Lines 214-247 in the revised manuscript).

We added the following sentences in the revised manuscript:

“Why are side-branching angles wider than bifurcation angles, and why do they vary more systematically with Tokunaga parameter c ? As Fig. 3b shows, junctions with larger differences k in Horton-Strahler orders are more common in networks with higher Tokunaga parameter c values. Junctions with larger order differences k , in turn, have larger contrasts in the slopes of their tributaries (as reflected in slope ratios farther below 1; Fig. 3c). This directly implies greater contrasts in channel slopes at side-branching junctions ($k > 0$) than at bifurcation junctions ($k = 0$), and greater channel slope contrasts in networks with greater degrees of side-branching (as measured by higher values of Tokunaga parameter c). Pairs of tributaries with larger differences in order tend to have

larger differences between their upstream slopes (i.e., smaller slope ratios SR) as a direct consequence of the power-law relationship between drainage area and channel slope⁵⁶, combined with Horton's exponential relationship between order and drainage area⁸. Consistent with Horton's steepest-descent geometric model⁸, tributaries with larger order differences k – and thus larger differences in their drainage areas and slopes – have wider average junction angles (Fig. 3d). This observation leads directly to the expectation that side-branching angles should be wider than bifurcation angles, and should be wider in networks with greater degrees of side-branching (larger values of Tokunaga parameter c).”

Please see Lines 214-231 in the revised manuscript.

“Slope ratios, and their corresponding junction angles, are topographic expressions of landscape dissection. A valley's high-order main stream will carry a relatively large discharge and therefore adjust toward a relatively low equilibrium channel gradient³⁴, whereas its side slopes will remain steep unless the streams draining those slopes have sufficient drainage areas (and thus discharges) to incise them⁷. Thus more feathered networks, whose tributary junctions will tend to have larger order differences k and greater contrasts in drainage areas and channel slopes, will also tend to have wider average junction angles. From this perspective, slope ratios and junction angles are not only a direct consequence of local erosion asymmetry, but also a broader result of side-branch formation, which ultimately sets a stream network's topology and geometry.” Please see Lines 238-247 in the revised manuscript.

Fig. 3. (in the revised manuscript) Effects of network topology, as expressed by Tokunaga parameter c , on junction angles between tributaries. Panel (a) shows how network-averaged side-branching angles (diamonds), bifurcation angles (circles), and all junction angles (triangles) vary with Tokunaga parameter c . Panel (b) shows how the proportions of junctions with each Horton-Strahler (H-S) order difference k vary with Tokunaga parameter c , averaged over the same bins as in (a). Boxplots (c) and (d)

show how average slope ratios between pairs of tributaries decrease (c) and junction angles widen (d) with increasing H-S order differences (k). In panels (b) and (d), boxes represent the interquartile range (IQR), with whiskers extending to 1.5 times the IQR. Outliers beyond this range are not shown.

These new analyses and expanded discussions mentioned above provide a physical explanation for the correlations reported in the original manuscript (L198-201, L271-274, L309-312, L317-317), as noted by the reviewer. We have further expanded the discussions of these relationships in the revised manuscript.

2. **See Lines 284-287 of the revised manuscript**

Lines 271-274 of the original manuscript said “*This observation suggests that the observed Spearman correlation between climate and network topology is primarily mediated through climate effects on topography and network geometry (Fig. 5) rather than through a direct impact of climate on network topology.*” To this we have added (**at Lines 284-287 in the revised manuscript**):

“This is consistent with the concept that climatic aridity affects channel incision, which in turn controls landscape dissection⁷, thereby setting Tokunaga parameter c , channel slopes and slope ratios, which ultimately impact side-branching angles.”

3. **See Lines 305-321 of the revised manuscript**

Lines 309-312 of the original manuscript said: “*This observation explains the strong partial correlation between side-branching angles and slope ratios ($\rho_{\text{partial}}=-0.43$; Fig. 5) and is consistent with the correlation between branching angles and slope ratios reported by Getraer and Maloof^{4,5}.*”

Lines 315-317 of the original manuscript said: “*While slope ratios strongly influence side-branching angles ($\rho_{\text{partial}}=-0.43$, $p<0.0001$), the effect of average channel slope is weaker ($\rho_{\text{partial}}=-0.13$, $p<0.0001$).*”

We expanded on the mechanistic framework (**see Lines 305-321 of the revised manuscript**):

“Stream networks delineate the pathways along which surface water aggregates, and their shapes and structures mirror the landscape’s development. Where erosional efficiency is high relative to diffusive smoothing, a branching instability⁷ triggers the growth of steep, low-order tributaries with small drainage areas (Fig. 1b). Viewed through this lens, Tokunaga parameter c expresses the prevalence of this side-branching instability: networks in more strongly dissected terrain exhibit higher c values and their junctions exhibit correspondingly stronger contrasts in stream orders, drainage areas (Supplementary Table S2), and channel slopes (Fig. 6). The prevalence of low-order side branches, which tend to run down valley walls perpendicular to the main stream^{8,43} (Fig. 1b), influences the statistics of network-averaged junction angles⁵⁰ and explains the strong partial correlation between network-averaged slope ratios and side-branching angles ($\rho_{\text{partial}}=-0.47$; Fig. 6). However, the effect of average channel slope on side-branching angles is weaker ($\rho_{\text{partial}}=-0.06$). Thus, these side-branching angles are more sensitive to how erosion creates

contrasts in slope between pairs of tributaries than to the corresponding average channel slopes (Fig. 6). In wetter climates, more effective fluvial incision produces stream junctions with greater contrasts in channel slopes (slope ratios farther below 1; $\rho_{\text{partial}}=-0.32$.”

4. See Figure 1 of the revised manuscript (or below)

Moreover, we added new panels to the revised Figure 1 to illustrate how stream networks with different topologies exhibit distinctive topographic embeddings, helping readers develop a conceptual link among network topology, geometry, and topography.

Fig. 1. (in the revised manuscript) Illustration of the Horton-Strahler and Tokunaga ordering systems, with example river networks. (a) The lines represent rivers within a 3rd-order stream network, differentiated by colors corresponding to their Horton-Strahler (H-S) orders. Each pair of Tokunaga orders (ω, ω') indicates a stream's H-S order (ω) and the order of the stream that it meets at its downstream end ($\omega' \geq \omega$). Junction types are defined by k , the (unsigned) difference in H-S orders between the two incoming tributaries: bifurcations occur when $k=0$, whereas side-branching junctions occur when $k>0$. (b) Three 5th-order networks with different Tokunaga parameter c values and examples of their topographic embedding. The locations of these three networks are indicated on the outline map.

Minor Comments

Manuscript structure:

The following is a suggestion which the authors can ignore if they choose: I think this manuscript would be easier to read (and thus more impactful) without using or minimizing the use of self-similarity and Takunaga's parameter c . After reading the manuscript, I believe the only benefit of including Takunaga's parameter c is to show that increases in the average side branching angle are a result of actual branching angles increasing and not a result of basins with inordinate amounts of large delta HS confluences (i.e., there are two ways to increase the average side

branching angle in a basin: have more large delta HS confluences relative to small delta HS confluences, or increase the branching angle of confluences). However, this same point, I think, could also be made in a different way which I believe will be more intuitive for most readers and will not take away the scientific rigor of the approach. Instead of using Tokunaga's parameter c as the x axis in Figure 3, I think these results could be broken up by delta HS values or potentially normalized by the average delta HS within the basin. This approach would clearly show the reader that increase in the average side-branching angles correlated with Aridity Index or average slope are a result of actual angles of the confluences increasing.

Response: We thank the reviewer for this thoughtful suggestion. Both Tokunaga parameter c and network-averaged Horton-Strahler order differences at junctions $\overline{\Delta HS}$ can serve as measures of network topology and are highly correlated with each other (Spearman $\rho=0.63$) among 11,946 5th-order Tokunaga self-similar stream networks analyzed in the main text (see Supplementary Text. S1).

While $\overline{\Delta HS}$ offers a simple, interpretable and direct measure of network topology, Tokunaga parameter c reflects the geometric progression of side-branch frequency with respect to order difference. As a result, Tokunaga parameter c is inherently scale-independent (see Supplementary Fig. S1), which is crucial for robust topological metrics. On the other hand, $\overline{\Delta HS}$ tends to increase with basin order (see Supplementary Fig. S1). Moreover, it is strongly influenced by the large number of low-order streams, which dominate most stream networks. As a result, $\overline{\Delta HS}$ may not fully capture the broader, scale-invariant properties of stream network topology (see Supplementary Text. S1), which is the reason why we opted for Tokunaga parameter c for our main analysis.

The revised supplementary material now includes a discussion and comparison of $\overline{\Delta HS}$ and Tokunaga parameter c (see Supplementary Text S1 and Fig. S1). As suggested by the reviewer, we also analyzed our dataset using $\overline{\Delta HS}$ (see Supplementary Fig. S4) instead of Tokunaga parameter c in Fig. 3 of the original manuscript (now Fig.4 of the revised manuscript) and obtained similar results (see Supplementary Fig. S4), further demonstrating the robustness and reliability of our main findings.

If the authors feel that Tokunaga's parameter c is still better, perhaps an alternative is to show both? One of the metrics could be shown in the supplement if space a concern.

Response: We thank the reviewer for this suggestion. For the reasons explained above, we kept Tokunaga parameter c for our main analysis. However, we expanded our analysis using $\overline{\Delta HS}$ in the revised supplementary information (see Supplementary Text S1, Figs. S1, S4 and our response above) to show that our results are robust with respect to different metrics of network topology.

Methods:

- Why was 10% of the network consisting of artificial channels chosen as a threshold cutoff? While a sensitivity analysis would be helpful to show the influence of artificial channels on the

results, I recognize this requires significant effort. If a sensitivity analysis is not performed in a future submission, it would be helpful to provide justification for the 10% cutoff threshold.

Response: We agree with the reviewer that the choice of 10% as a cutoff to minimize artificial effects from engineered channels, such as canals and ditches, is somewhat arbitrary.

A small proportion of artificial channels is common, especially in low-relief regions, and using a stricter threshold would disproportionately exclude basins in these landscapes. The 10% threshold, therefore, balances the need to remove networks with pervasive human disturbance while retaining basins whose artificial channels are limited and spatially localized. In addition, we now provide results using alternative 5% and 20% thresholds in the supporting information, which yield similar results (see **Supplementary Tables S3 and S4**), demonstrating that our findings are robust to the choice of cutoff. We have added this information to the revised methods: *“The 10% threshold removes networks with pervasive human disturbance while retaining basins whose artificial channels are limited and spatially localized. Using a threshold of 5% or 20% instead of 10% yields similar results (Supplementary Tables S3-S4).”* **Please see Lines 372-375 in the revised manuscript.**

- More information about the lack of slope information connected with certain stream segments would be helpful for the reader. How many stream segments lack slope information? Is there a spatial trend to the lack of slope information? Similar to the previous comments, how does this impact the results?

Response: Among the 11,946 5th-order Tokunaga self-similar networks included in our main analysis, only 0.1% of flowlines lack slope information. We now report this information in the revised manuscript: *“To calculate the slope of each segment, we exclude any flowlines lacking slope data (~0.1% of all the flowlines in 11,946 5th-order networks analyzed in the main text).”* **Please see Line 379-381 in the revised manuscript.** These missing values occur primarily in flat regions, where the digital elevation model provides insufficient vertical contrast for reliable slope estimation. Given their tiny proportion, the absence of slope information has a negligible effect on the overall statistical analyses and conclusions.

Note that in the revised manuscript, in response to Reviewer #4’s concern regarding channel slopes, we additionally excluded stream networks in which all the flowlines have a slope of 0.00001 (see **Lines 368-369 of the revised manuscript**), which is the slope cutoff limit in the NHDPlus-HR dataset. As a result, the total number of 5th-order stream networks decreased from 17,021 in the original manuscript to 16,322 in the revised analysis. The revised analysis yields results that are consistent with those of the original manuscript, demonstrating the robustness of our findings.

Line and Figure Comments

- Line 52-56: It appears the variable k , may represent two things. Text on line 52 implies k may be the order of the tributary, while Lines 58-59 clearly define k as the difference in Strahler stream orders. Can you please clarify this text?

Response: We thank the reviewer for pointing out this ambiguity. We have clarified throughout the revised manuscript that k denotes the difference in Horton-Strahler order at a junction between two tributaries. Particularly, we have revised the respective sentence to: “*To quantify stream network topology (the connectivity between streams of different orders), Tokunaga^{22,23} expanded the Horton-Strahler ordering scheme by introducing the concept of side branches (streams of order ω entering streams of higher order ω' , with order differences $k=\omega'-\omega$, $k>0$), and bifurcations (streams of the same order merging at a junction, with order difference $k=0$; see Fig. 1a).*”

Please see Lines 50-54 in the revised manuscript. We have also checked the entire manuscript to ensure that the definition and usage of k are consistent throughout.

- Line 84-86: How are channel heads identified if a threshold drainage area is not used?

Response: As channel heads, we refer to the starting points of the first-order channels as mapped by USGS in the NHDHR Plus dataset. Channel heads have been identified by USGS either through local mapping or regional thresholds/curvature measures. Please see the NHDPlus-HR User Guide for more details.

Reference:

Moore, R.B., McKay, L.D., Rea, A.H., Bondelid, T.R., Price, C.V., Dewald, T.G., Hayes, L., 2025. User’s guide for the National Hydrography Dataset Plus High Resolution (NHDPlus HR) (No. 2025–5031), Scientific Investigations Report. U.S. Geological Survey. <https://doi.org/10.3133/sir20255031>

- Line 87-90: I would suggest either softening this language or describing the exact question that is answered here in more detail. Many workers have done substantial work on this topic, and many questions are left. Please give citations for the long-standing questions as well.

Response: We thank the reviewer for their suggestion and have now revised the text to soften the language and clarify the specific questions addressed in this study, and we cite the relevant foundational work. Line 87-90 of the original manuscript said: “*As a result, it more accurately reflects natural river networks and avoids artifacts caused by assuming constant drainage areas at channel heads³⁹. With this dataset in hand, our analysis solves long-standing questions about the degree of self-similar scaling in real-world river networks and about what influences their topology.*” Now the revised sentences are: “*Testing these findings in a natural system requires stream networks mapped at high resolution across a wide range of climatic conditions. Here we analyze stream networks from the high-resolution National Hydrographic Dataset⁴⁰ (NHDPlus-HR), which is extensively ground-checked and provides the best available continental-scale mapped stream networks⁴¹. This allows us to re-examine classic questions about how much real-world stream networks exhibit self-similar scaling, and how topography and climate influence their topology^{24,25,27,28,29,30,38,42}.*” **Please see Lines 83-90 in the revised manuscript.**

- Line 96: Please ensure at least one of these citations is appropriate support for the topology and geometry argument in this sentence. From my recollection the cited references mainly discuss the morphologic characteristics, but I could be wrong.

Response: We thank the reviewer for raising this point. Upon revision, we streamlined the Introduction to improve clarity and focus, and revised the sentence to: “*While network topology describes the connectivity of streams of different orders, it does not capture the network’s geometry—namely, the length of segments⁵ and the angles between them^{8,43,44,45,46,47,48,49,50,51,52}”.*

Please see Lines 101-103 in the revised manuscript. The cited references at Line 96 of the original manuscript mainly address landscape equilibrium and drainage divides and are no longer directly relevant to the revised sentence. Therefore, they are no longer cited.

- Line 121: More information would be helpful to let the reader know why a two-step approach is necessary here. It is unclear how a river can be self-similar but not meet Tokunaga’s scaling criteria.

Response: We thank the reviewer for this question. Tokunaga self-similarity is a special kind of self-similarity constraint by the exponential regularity in the side-branch structure of hierarchical networks, expressed through Tokunaga parameters (Equation 1 in the revised manuscript; see Line 58 in the revised manuscript).

Thus, testing Tokunaga self-similarity requires a first test for self-similarity, which can be tested using ANOVA (Zanardo et al., 2013). Then the self-similar network can be tested to see whether it follows Tokunaga’s scaling law (Equation 1 in the revised manuscript; Zanardo et al., 2013).

To reduce confusion for readers, we have streamlined the text in the “*Test of stream self-similarity*” section of the main text and revised related sentences to: “*Because Tokunaga parameter c assumes topological self-similarity^{23,24}, we limit our analysis to Tokunaga self-similar networks. Of the 16,332 5th-order stream networks from the NHDPlus-HR dataset, 73% (11,946) pass the Tokunaga self-similarity criteria of ref. 24 (see Supplementary Text S2 and Fig. S2).*” **Please see Lines 126-129 of the revised manuscript.**

Additionally, we moved much of the technical details for testing Tokunaga self-similarity to the supplementary materials (see **Supplementary Text S2**). The reason to do so is twofold. First, much of the description is based on previous literature, mostly following the procedures of (Zanardo et al., 2013). Second, not being limited by word count constraints allowed us to explain the procedures in more detail in the supplementary information, making it easier for an interested reader to follow the different calculation steps.

Reference:

Zanardo, S., Zaliapin, I. & Foufoula-Georgiou, E. Are American rivers Tokunaga self-similar? New results on fluvial network topology and its climatic dependence. *J. Geophys. Res. Earth Surf.* 118, 166–183 (2013).

- Line 131: Although the slope ratio is well described in the methods, SR is a major component of this story and, for the reader to understand the results described here, a better main text explanation of SR would be helpful.

Response: We thank the reviewer for this suggestion. To improve clarity in the main text, we have expanded the description of slope ratio (SR) in the *Results and Discussion* section (see Lines 133-135 in the revised manuscript): “Here, slope ratio is defined as the ratio of the gentler to the steeper slope and therefore ranges from 0 to 1 (see Methods). \overline{SR} values closer to 1 indicate smaller contrasts in slopes between pairs of tributaries.”

We have also clarified the implications of our results regarding SR and the proportion of Tokunaga self-similar networks early in “*Test of stream network self-similarity*” of *Results and Discussion* section (see Lines 140-143 in the revised manuscript), and further expanded the discussions of the relationships among slope ratio, network topology and branching geometry later in “*Interrelations of stream network topology and planform geometry*” of *Results and Discussion* section (see Lines 214-247 in the revised manuscript), as suggested by the reviewer.

Additionally, we have added a sentence to the revised introduction to provide a broader context for its geomorphic significance: “While stream branching angles are only weakly correlated with the average slopes of the two tributaries⁴⁷, they are more strongly correlated with the contrast in slopes at the confluence⁴⁹. According to Horton’s geometric model⁸, branching angles depend on the ratio between the slopes of the shallower stream and the steeper one (the slope ratio SR).”

Please see Lines 106-110 in the revised manuscript.

- Line 144-146: If c is kept in the manuscript, I suggest moving this description of c to earlier in the text. The variable is referred to frequently and a description such as this helps readers to conceptualize the variable.

Response: We thank the reviewer for this suggestion and have now moved the sentence to the introduction of the revised manuscript: “As illustrated in Fig. 1b, larger values of parameter c imply a greater abundance of low-order channels joining high-order streams, resulting in a more ‘feathered’ network. While parameter a varies only slightly across different networks, parameter c is sensitive to climatic influence²⁴.” Please see Lines 66-69 in the revised manuscript. Additionally, the example networks with different Tokunaga parameter c values shown in Fig. 1b of the revised manuscript will further help readers to conceptualize the variable.

- Line 152: Can you provide a citation?

Response: Yes, we revised sentence to: “Individual hexagons with less feathered networks are found along coastlines and in the glaciated upper Midwest (Fig. 2a), presumably reflecting local geomorphic controls⁴².” Please see Lines 157-159 in the revised manuscript.

Reference:

Danesh-Yazdi, M., Tejedor, A. & Foufoula-Georgiou, E. Self-dissimilar landscapes: Revealing the signature of geologic constraints on landscape dissection via topologic and multi-scale analysis. *Geomorphology* **295**, 16–27 (2017).

- Line 180-182: I suggest moving this description of side branching earlier. It will help the reader understand the arguments presented before this point.

Response: We agree with the reviewer and have now described side branching in the introduction of the revised manuscript: “*To quantify stream network topology (the connectivity between streams of different orders), Tokunaga^{22,23} expanded the Horton-Strahler ordering scheme by introducing the concept of side branches (streams of order ω entering streams of higher order ω' , with order differences $k=\omega'-\omega$, $k>0$), and bifurcations (streams of the same order merging at a junction, with order difference $k=0$; see Fig. 1a)*”. Please see Lines 50-54 in the revised manuscript.

- Line 320: I do not believe the analysis provided in this manuscript provides support for this statement. If the authors would like to argue that side-branching angles are not controlled by tectonics, a more robust analysis of this is required. I would suggest removing the text “overall rates of uplift and” and change steepness to slope. The word steepness suggests the normalized metric, channels steepness, was used in this analysis.

Response: We thank the reviewer for this suggestion. Accordingly, we have revised the sentence to: “*Thus, these side-branching angles are more sensitive to how erosion creates contrasts in slope between pairs of tributaries than to the corresponding average channel slopes (Fig. 6).*” to make this point clearer. Please see Lines 317-319 in the revised manuscript.

- Figure 3: Having the same symbols in panel b represent something different than they do in panels c-f is confusing for the reader. Please change.

Response: Good point. To avoid potential confusion, we have split Figure 3 in the original manuscript into Figures 3 and 4 in the revised manuscript, as additional results have been added to the revised Figure 3. Following the reviewer’s suggestion, we have modified the symbols so that those used in panel b of the original Figure 3 (Panel a of the revised Figure 3) are different from those used in panels c-f of the original Figure 3 (panels a-d of the revised Figure 4).

Remarks on code availability:

I only very briefly looked at the code. The readme file seemed to be empty and did not have instructions. I saw code was present in the other file, but I did not inspect it.

Response: We thank the reviewer for pointing this out. We have now added a detailed README that includes clear code instructions and step-by-step instructions for reproducing the analyses and figures presented in the paper. Please see https://github.com/MhL-2024/Network_Topology. More details regarding the datasets used in the codes are available at https://yaleedu-my.sharepoint.com/:f:/g/personal/minhui_li_yale_edu/IgASI3ZhskBTQqlB_nuDuVFfAdp4STY

[wZvTyTFDkBRdfIQc?e=SX1K3J](https://github.com/MhL-2024/Network_Topology) for review purpose. The data will be posted to a FAIR repository with a permanent DOI upon acceptance.

Reviewer #2

Remarks on code availability:

Code appears to produce plots shown in manuscript but does not contain the more detailed analyses of channel networks.

Response: We thank the reviewer for this careful observation. In the revised version, we have expanded the repository to include:

(1) scripts performing the core analyses of stream network topology and branching geometry and reproducing the figures in the main text (see https://github.com/MhL-2024/Network_Topology);
(2) links to all the processed datasets used in our analyses: https://yaleedu-my.sharepoint.com/:f/g/personal/minhui_li_yale_edu/IgASI3ZhskBTQqlB_nuDuvVfAdp4STY_wZvTyTFDkBRdfIQc?e=SX1K3J, which are currently hosted on OneDrive for review purposes and will be made openly accessible via a permanent public repository upon acceptance.

Reviewer #3

This manuscript sets out a compelling data-driven argument for how climatic signature is expressed in river network topology – not directly, but through mediating conditions of basin topography and network geometry.

The analysis draws upon a collectively extensive body of work that has explored each of these contributing components – topography, network geometry, and network topology – respectively in relation to climatic forcing without quite disentangling the nature of their inter-relationship. This contribution gets closer to that aim, leveraging both a high-resolution dataset of channel networks for the contiguous USA, and a clever application of partial rank correlation statistics to unpack subtle but significant indirect links through which climatic forcing is imbued in network topology.

My remarks on this manuscript are relatively minor, but pertain to the narrative. I hope my comments might help the authors clarify their message that much more, for their readers' benefit.

Response: We greatly appreciate the recognition of our study's contribution in clarifying how climatic influences on river network topology are mediated through topography and network geometry. We also thank the reviewer for their insightful suggestions, which have helped us

improve the clarity and coherence of the manuscript. Detailed point-by-point responses to all comments are provided below.

The first sentence of the final paragraph (L327–329) presents the clearest summary framing (problem and findings) in the manuscript – so much so that I think it should get worked into the introduction.

Response: We thank the reviewer for these suggestions. L327-329 of the original manuscript said, “*Ultimately, our observations suggest that climatic aridity may shape both basin topography (i.e., mean channel slope, slope ratios) and network geometry (i.e., side-branching angles), and therefore indirectly influence network topology (i.e., Tokunaga parameter c).*” We agree with the reviewer that much of a network’s topology and geometry is a consequence of its embedding in the three-dimensional topography, and this perspective indeed provides a clear conceptual framing of the study.

Because this statement synthesizes the main findings of our analysis, we felt that introducing it in the introduction (before presenting the results) might be premature. Instead, we explicitly state this point now at the end of the *Abstract*, where the core message of the study is appropriately summarized: “*These findings demonstrate the co-evolution of network geometry, topography, and topology under the influence of landscape evolution driven by climatic forcing.*” **Please see Lines 30-32 in the revised manuscript.**

To me, the introductory paragraphs dive too quickly into how topology and topography are quantified, and there are some intermediate punchlines (e.g., L88–89) that are confusing. (I understand that the NHDPlus-HR dataset allows previously inaccessible insight into self-similarity in real river networks, and therefore robust application of Tokunaga ordering. The authors should absolutely point this out – but as written it diverts the reader from the larger aim; it's an important finding, but also a means to an end.)

Response: We thank the reviewer for the suggestions. We retain the description of how we quantify topology in the introduction section of the revised manuscript because the Tokunaga parameters are central to our analysis, and introducing them early helps orient readers to the main methodological and conceptual framework, as suggested by reviewer #1. Technical details for testing Tokunaga self-similarity are now moved to the **Supplementary Text S2**, as this allows a more extensive description of the calculation without distracting the reader from the main goals of the paper. We have strengthened the statement of the key research gaps and scientific questions in the introduction section to ensure the narrative does not divert from the manuscript’s primary goal.

Relevant sentences in the introduction of the revised manuscript are:

- (1) “*Here, we analyze the relationships between stream network topology and geometry, and explore how climate influences these properties through the networks’ embedding in Earth’s three-dimensional topography.*” **Please see Lines 46-49 in the revised manuscript.**

- (2) “Yet, estimates of this climatic sensitivity vary with the area thresholds used to extract topographic flow paths²⁴, and whether such climatic sensitivity persists at the continental scale remains unresolved.” Please see Lines 69-71 in the revised manuscript.
- (3) “Previous research on stream network topology has mainly focused on determining whether principles such as topological randomness²⁵ or optimality^{6,14,26} can generate realistic-looking network configurations^{27,28,29,30}. However, these statistical methods have largely ignored the topographic controls that actively shape network topology^{7,31}.” Please see Lines 72-75 in the revised manuscript.
- (4) “Despite these insights, it remains unclear how climatically mediated landscape dissection influences stream network branching geometry at side-branches and bifurcations, as well as network topology.” Please see Lines 112-115 in the revised manuscript.

Regarding L88-89 in the original manuscript, it said, “With this dataset in hand, our analysis solves long-standing questions about the degree of self-similar scaling in real-world river networks and about what influences their topology.” Following the suggestions of both Reviewer #1, Reviewer #3 and Reviewer #4, we have revised it to: “Here we analyze stream networks from the high-resolution National Hydrographic Dataset⁴⁰ (NHDPlus-HR), which is extensively ground-checked and provides the best available continental-scale mapped stream networks⁴¹. This allows us to re-examine classic questions about how much real-world stream networks exhibit self-similar scaling, and how topography and climate influence their topology^{24,25,27,28,29,30,38,42}.” Please see Lines 85-90 in the revised manuscript.

I wonder if the authors might shift the paragraph at L106 up to be the second paragraph (inset at L49). The paragraph on Tokunaga ordering (L49) could introduce the Results section at L117. For continuity, I would also put the paragraph at L141 above the section break, to be the final paragraph of the first Results section – that is, insert after L139. Similarly, the paragraph on topography (L91) sets up the Results section at L147, and could serve as the opening paragraph there.

Response: We thank the reviewer for these thoughtful structural suggestions. We carefully considered the proposed reordering. In the revised manuscript, we retain the paragraph at L106 from the initial manuscript because it serves as a summary transition after introducing network topology, geometry, and the NHDPlus-HR dataset. We have revised the paragraph to clarify its role and improve the narrative flow:

“To better understand how the planform geometry and topological connectivity of stream networks are embedded in three-dimensional landscapes, we first test Tokunaga self-similarity in 16,322 5th-order real-world stream networks across the contiguous United States using the high-resolution National Hydrographic Dataset⁴⁰. We then analyze relationships among Tokunaga parameter c , bifurcation and side-branching angles, slope ratios, and climatic aridity. This information allows us to develop a conceptual framework that explains how climate influences network topology

through its impact on topography and network geometry.” Please see Lines 116-123 in the revised manuscript.

We have retained and revised the paragraph on Tokunaga ordering (L49 of the original manuscript) in the introduction (see Lines 50-71 in the revised manuscript). Because the Tokunaga parameters are central to our analysis, introducing them early helps orient readers to the main methodological and conceptual framework, as suggested by Reviewer #1.

While the reviewer recommended moving the paragraph at Line 141 of the original manuscript to the end of the first Results section, we instead placed it at the beginning of this section to provide clearer context for the results that follow. To further streamline the text, we have revised the Results section at L117 of the original manuscript (see Lines 126-129 in the revised manuscript) to focus on the key findings: “*Because Tokunaga parameter c assumes topological self-similarity^{23,24}, we limit our analysis to Tokunaga self-similar networks. Of the 16,332 5th-order stream networks from the NHDPlus-HR dataset, 73% (11,946) pass the Tokunaga self-similarity criteria of ref.24 (see Supplementary Text S2 and Fig. S2).*”. Additionally, we have moved more detailed and highly technical methodological descriptions to the revised **Supplementary Text S2**.

Following the reviewer’s suggestion, the sentence at L91 of the original manuscript has been moved to serve as the opening sentence in the result section at L147 of the original manuscript: “*Stream network branching patterns result from the evolution of the landscape they are embedded in, and thus can be used to infer climatic and tectonic factors that shape the landscape^{7,32,33,36}.*” Please see Lines 150-152 in the revised manuscript.

I really enjoyed the synthesis section from L252 onward, and I think it's one of the strengths of this manuscript. In that vein, I encourage the authors, when they re-read this work with fresh eyes, to do their best to make each paragraph a ratchet that moves the reader inexorably forward. (Put another way, be on the lookout for recursions in the text.)

Response: We thank the reviewer for the positive feedback and for highlighting the synthesis section as a strength of the manuscript. In the revised manuscript, we expanded the physical explanations for our results and carefully re-read the synthesis section and the broader discussion to streamline paragraph transitions and strengthen the coherence and forward momentum of the narrative. Please see Lines 267-342 in the revised manuscript.

I look forward to seeing this work in print – congratulations to the authors on a fine analysis.

Response: We sincerely thank the reviewer for the encouraging words and for their thoughtful feedback throughout.

Remarks on code availability:

N/A

Reviewer #4

This paper looks at relationships between networks topology, channel gradients, lithology and climate. The paper shows that gradients are more highly correlated with a metric called the Tokunaga ratio than they are with climate. I have a number of questions related to the analysis but overall I think this is a very interesting paper and shows something new. One of the most interesting aspects of this work is that the Tokunaga ratio is correlated with the side branching angle and not the bifurcation angle, suggesting that the process of adding smaller links leads to different channel geometry than bifurcation. To me this has interesting implications for network growth, has never been shown before, and somehow the authors have not deemed it important enough to mention in the abstract. I would change that.

Response: We thank the reviewer for the positive feedback and for highlighting the important implications of the relationship between the Tokunaga parameter and side-branching angles for network growth. Following this helpful suggestion, we now emphasize this point more clearly in the revised manuscript. Specifically, we have added the sentence “*small tributaries join larger streams at systematically wider angles*” in the revised abstract (see Lines 25-26 in the revised manuscript). We also added a sentence in the result section: “*The patterns in Fig. 5 imply that processes associated with the creation of low-order side-branches, such as lateral erosion⁵⁷ may produce junction geometries that differ greatly from those generated by bifurcation⁵⁰.*” Please see Lines 254-256 in the revised manuscript.

I think the choice of using 5th order basins vs 6th or 7th order could be clearer. On the other end of the size spectrum, I think a comment about how many 5th order basins cross big physiographic divides might be in order (that is, does the basin size lead to many basins crossing from steep mountains into gentle basins). The reason for a comment on this is the somewhat strange pattern in the slope map (see comment below).

Response: We thank the reviewer for this helpful suggestion. We have clarified the rationale for selecting 5th-order basins in the supplementary information (see Supplementary Text S3). This order was chosen for the primary analysis as a balance between spatial resolution and topological completeness: basins of this topological size are large enough to encompass well-developed hierarchical structures (and thus allow robust estimation of Tokunaga statistics) yet small enough to maintain a clear correspondence with local climatic and topographic conditions. Using 6th- or higher-order networks would substantially reduce the number of independent samples and increase internal heterogeneity, thereby weakening statistical power and interpretability. To explore scale effects, we have included analyses for 6th-order basins in the supplementary information (see Supplementary Text S3 and Table S1), which yield similar results and confirm the robustness of our findings. We have reported these results also in the main text: “*To explore how varying stream network scales could affect our results, we also repeated our main analysis on the 3,454 6th-order stream networks in the contiguous U.S., resulting in similar conclusions (Supplementary Text S3 and Table S1).*” Please see Lines 145-148 in the revised manuscript.

To mitigate potential artifacts arising from larger basins crossing major physiographic boundaries, we have updated the slope map to display the *median* (rather than *mean*) slope within each hexagon, thereby reducing the influence of extreme values in transition zones. We have also excluded networks in which all flowlines have a slope of 0.00001 due to the slope cutoff in the NHDPlus-HR dataset in the flat regions. As a result, the total number of 5th-order stream networks decreases from 17,021 in the original manuscript to 16,322 in the revised manuscript. The revised analysis yields results that are consistent with those of the original manuscript, demonstrating the robustness of our findings. Additional discussion of this issue is provided in the detailed response below.

Overall I think this is a very interesting contribution, raises some new questions about network topology, and I would characterise most of my comments as cosmetic.

Response: We sincerely thank the reviewer for the positive evaluation and the constructive suggestions, which have helped us further improve the clarity and presentation of the manuscript.

Lined comments

Line 64: You later say that you cannot reproduce this result (or at least you think climate is of secondary importance). The way this is written seems (to me at least) that you are taking this result as a given, when in fact there is only one paper that says this and that your work does not support it.

Response: We fully agree with the reviewer and revised the sentence to “*While parameter a varies only slightly across different networks, parameter c is sensitive to climatic influences²⁴. Yet, estimates of this climatic sensitivity vary with the area thresholds used to extract topographic flow paths²⁴, and whether such climatic sensitivity persists at the continental scale remains unresolved.*”

Please see Lines 68-71 in the revised manuscript.

Figure 1: At this point in the paper ω is not defined. It should be somewhere (perhaps above equation 1). Also the labelling is inconsistent (are you labelling nodes or edges, and if one or the other why aren't all of them labelled?).

Response: We thank the reviewer for pointing out this omission. We have now defined ω both above equation 1 in the text and in the Figure caption as the Strahler order of the side branch, while ω' denotes the order of the stream that it joins. The revised sentence now reads as “*To quantify stream network topology (the connectivity between streams of different orders), Tokunaga^{22,23} expanded the Horton-Strahler ordering scheme by introducing the concept of side branches (streams of order ω entering streams of higher order ω' , with order differences $k=\omega'-\omega$, $k>0$), and bifurcations (streams of the same order merging at a junction, with order difference $k=0$; see Fig. 1a).*” (see Lines 50-54 of the revised manuscript) and “*Each pair of Tokunaga orders (ω , ω') indicates a stream's H-S order (ω) and the order of the stream that it meets at its downstream end ($\omega \geq \omega'$).*” (in the caption of Figure 1, see Lines 94-96 of the revised manuscript).

Regarding the labelling in Figure 1, the Tokunaga order is assigned to the entire stream of a given Horton–Strahler order. A side-branching tributary may split this stream into multiple pieces, but these pieces collectively represent a single Tokunaga interaction and therefore share the same Tokunaga order. This is indicated in the example river networks of Figure 1a through the stream's colors.

Line 81-90: The novelty of this paper (and I think it is quite novel) is the analysis. It is not the underlying data. These lines have many problems and are not necessary for the manuscript.

Response: We thank the reviewer for this helpful clarification and for recognizing the novelty of our analytical approach. We fully agree that the paper's strength lies in its analysis and interpretation rather than in the underlying datasets. Accordingly, we have streamlined the related section: *“Testing these findings in a natural system requires stream networks mapped at high resolution across a wide range of climatic conditions. Here we analyze stream networks from the high-resolution National Hydrographic Dataset⁴⁰(NHDPlus-HR), which is extensively ground-checked and provides the best available continental-scale mapped stream networks⁴¹. This allows us to re-examine classic questions about how much real-world stream networks exhibit self-similar scaling, and how topography and climate influence their topology^{24,25,27,28,29,30,38,42}.”* **Please see Lines 83-90 in the revised manuscript.**

The report cited here has been superseded by a 2025 report (Moore et al., 2025). It does not really explain how channel heads are found in the NHDPlus HR dataset. It does explain the many corrections needed to be made to the underlying NHD data because once the lidar was collected it emerged that many of the channels mapped on USGS quads were not actually in the right place (see page 60 of the report). So the NHDPlus HR is already corrected by topographic data. The authors then say that other datasets use a threshold drainage area, which leads to inconsistent channel networks. But as the report from Terziotti and Archuleta (2020) makes clear, the NHD is also inconsistent (from their page 16):

“The original NHD was digitized from individual 7.5-minute quadrangle map sheets that were compiled at different times, by many individuals, using varied sources; therefore, some areas of the country have hydrography that is represented at different densities. These discrepancies are due to differing source material or standards and procedures and are not due to differences in geomorphology or hydrologic conditions.”

This report goes on to explain how these differences in channel heads would be fixed, and advocates for a variety of methods for repositioning channel heads. They will insert “additional features” (including new channel heads)

- “● if there is clear evidence of the feature in the elevation data source,
- if there is clear evidence of the feature using an appropriate ancillary data source,

- if a method has given good results for delineation of stream channels or other features, and it is quality assured using the elevation data and other high-quality ancillary datasets”

So the overall impression is that the NHDPlus HR is hand curated in some way that is not reproducible and is just as susceptible to bias as an algorithmically-derived channel network. The user guide of Moore et al (2025) doesn't explain how this is done so we are left to guess (unlike networks that use a threshold drainage network and the threshold drainage area is reported). I have a strong suspicion the results using a threshold extraction or a more advanced channel head finding algorithm on the NDEP lidar data would result in outcomes indistinguishable from those reported here, NHDPlus HR is as good a network as any (and if you think hand curation is better, it might be better), and I don't think there needs to be any modification to the analysis. But this paragraph makes misleading statements and needs to go. Just say “The channel network used was the NHDPlus HR dataset”.

Moore, R.B., McKay, L.D., Rea, A.H., Bondelid, T.R., Price, C.V., Dewald, T.G., Hayes, L., 2025. User's guide for the National Hydrography Dataset Plus High Resolution (NHDPlus HR) (No. 2025–5031), Scientific Investigations Report. U.S. Geological Survey. <https://doi.org/10.3133/sir20255031>

Terziotti, S., Archuleta, C.-A., 2020. Elevation-derived hydrography acquisition specifications (No. 11-B11), Techniques and Methods. U.S. Geological Survey. <https://doi.org/10.3133/tm11B11>

Response: We sincerely thank the reviewer for drawing our attention to the user guide of Moore et al. (2025) and for the detailed explanation regarding the NHDPlus-HR dataset. We have revised our manuscript based on the suggestions. Please see Lines 83-90 in the revised manuscript (as quoted in response to the reviewer's directly preceding comment).

Line 108: Why 5th order? In 5th order channels, when you do the regression to calculate c you are only fitting 4 points. Do your results change with 6th order basins, for example?

Response: We thank the reviewer for this thoughtful question. As mentioned above, we chose 5th-order basins to ensure a sufficient number of independent samples that cover diverse landscapes across the contiguous U.S. while maintaining adequate internal network structure for estimating Tokunaga parameters. We have discussed this in **Supplementary Text S3**.

To evaluate sensitivity to basin order, we have repeated our main analysis using 6th-order basins, which yield similar results. We have reported this in the revised manuscript (see Lines 145-148 in the revised manuscript), with more details in **Supplementary Text S3 and Table S1**.

Line 120-122: The explanation of the ANOVA test is missing from the methods. I don't think it will change the result, but why was a non-parametric test not chosen for this step? Zanardo et al have an entire paragraph about the drawbacks of the ANOVA (paragraph 31 in their paper) and I never understood why they didn't use a non-parametric test.

Response: We thank the reviewer for this insightful comment. In Zanardo et al. (2013), the reliability of the ANOVA test was evaluated through Monte Carlo simulations. They generated large ensembles of synthetic networks to estimate the empirical significance level associated with

the ANOVA statistics, finding that a nominal level of $\beta=0.0052$ corresponds to a 5% false-rejection rate of the true null hypothesis.

To further verify robustness, we additionally applied a non-parametric Kruskal-Wallis test instead of ANOVA. We obtained very similar self-similar rates (0.9 vs. 0.86 at a significance level of $p=0.05$), confirming that our results are not sensitive to the choice of the statistical test. We now show these results in the Supplementary Text S2.

Line 130-132: Climate varies over a longer wavelength (loosely defined) than lithology. That is, in a 5th order channel network you are unlikely to have a large variation in the aridity index, but you could have several rock types with very different properties. The rock hardness is quite strongly correlated to the various slope metrics. The lithology is assigned of 50% of the underlying rocks are the same in a catchment, but harder lithologies in the headwaters and softer lithologies downstream will have quite a different effect, I imagine, on topology than vice versa. Some comment on this would be welcome.

Response: We thank the reviewer for this excellent point, and we agree that lithologic heterogeneity can influence network topology in ways that differ from broader, smooth climatic gradients. However, we didn't observe a significant influence of lithology on the fraction of self-similar stream networks. We acknowledge that our lithology assignment, based on the dominant (>50%) rock type within each catchment, represents a simplification that may mask sub-basin heterogeneity. The lithology dataset (Hartmann & Moosdorf) available for the continental US, with its relatively coarse spatial resolution ($\sim 0.5^\circ$), restricts our ability to fully capture the effects of lithologic variability at finer scales.

Reference:

Hartmann, J. & Moosdorf, N. The new global lithological map database GLiM: A representation of rock properties at the Earth surface. *Geochem. Geophys. Geosyst.* **13**, 12 (2012).

Line 130-131 and Line 371-373: I find it counterintuitive to select a slope ratio, which is a metric specifically designed to test differences in gradients in streams of different order, in such a way that a larger value represents a smaller variation in channel gradients. Surely this metric should be chosen so a higher slope ratio represents a larger variation in channel gradients.

Response: We thank the reviewer for this comment. We agree that this formulation of the slope ratio is counterintuitive, but it follows the classical Hortonian convention, such that slope ratios closer to 1 indicate similar gradients between the merging streams, whereas smaller ratios (farther below 1) reflect large slope contrasts. This is also the slope ratio convention used by Getraer and Maloof (ref. 49) in their recent work on this topic. We have clarified the definition in the revised manuscript: “According to Horton’s geometric model⁸, branching angles depend on the ratio between the slopes of the shallower stream and the steeper one (the slope ratio SR).” (see Lines 108-110 of the revised manuscript) and “Here, slope ratio is defined as the ratio of the gentler to the steeper slope and therefore ranges from 0 to 1 (see Methods). \overline{SR} values closer to 1 indicate

smaller contrasts in slopes between pairs of tributaries.” (see Lines 133-135 of the revised manuscript). More detailed descriptions of slope ratio are shown in the Methods. Please see Lines 382-385 in the revised manuscript.

We also revised the text throughout the manuscript to clearly indicate that slope ratios close to 1 denote smaller slope contrasts. Please see Lines 134-135, 143, 218-219, 223 in the revised manuscript.

Lines 143-149: Can you please report the distribution of sizes of the basins. How many are in each 10,000km² hex?

Response: We thank the reviewer for this question. We have now included such information in the revised manuscript:” *The 11,946 5th-order Tokunaga self-similar networks have drainage areas with 20th, 50th, and 80th percentiles of 21 km², 90 km², and 297 km², respectively.*” (see Lines 144-145 in the revised manuscript) and “*Each hexagon contains, on average, thirteen 5th-order Tokunaga self-similar networks.*” Please see Lines 155-156 in the revised manuscript.

Line 152: You mention here clustering of more feathered networks in specific geomorphic settings, and then say this “presumably reflects geological controls”. If you think it is glaciers causing the changing topology then you are implying it is **not** a geological control.

Response: We thank the reviewer for pointing out this inconsistency. We have revised “geological controls” to “geomorphic controls”. The revised sentence now reads: “*Individual hexagons with less feathered networks are found along coastlines and in the glaciated upper Midwest (Fig. 2a), presumably reflecting local geomorphic controls⁴².*” Please see Lines 157-159 in the revised manuscript.

Figure 2f: I presume, because you are using NHDPlus, the more arid catchments have a lower drainage density. The slope ratio between two parts of a drainage network depends on the ratio of drainage area raised to the power of the concavity index. We know there is a climate control on the concavity index. Is the signal in panel f controlled by this or by the difference in area ratios because of the changing drainage densities?

Response: Figure 2f of the original manuscript showed relationships between binned mean values of Tokunaga parameter c and network-averaged slope ratio. In the revised manuscript, we have now added new results (Fig.3) and expanded the physical interpretation linking Tokunaga parameter c , slope ratio, side-branching angle and bifurcation angle (see Lines 193-247 in the revised manuscript), as suggested by Reviewer #1.

The reviewer’s question appears to be whether parameter c varies with AI because of

causal chain #1: AI->drainage density->(area ratio and slope ratio)->parameter c ,

or because of **causal chain #2:** AI -> concavity -> slope ratio -> parameter c .

We don't think either of these causal chains provides a clear explanation for our results.

We can explore **causal chain #1** by looking at the relevant correlations. Our analysis of 11,946 5th-order Tokunaga self-similar networks indicates that AI is more strongly related to area ratio (Spearman $\rho=-0.26$, $p<0.0001$) and slope ratio (Spearman $\rho=-0.44$, $p<0.0001$) than to drainage density (Spearman $\rho=0.17$, $p<0.0001$). This suggests that the major effect of climate is on the degree of dissection (how incised channels are, which creates opportunities for steeper tributary channels and wider branching angles), rather than large changes in drainage density.

To explore **causal chain #2**, we calculated a concavity index (θ) for each network, fitting the scaling relation $S \sim A^{-\theta}$ where S is the local flowline slope and A is upstream drainage area. Slope and drainage area were obtained from the NHDPlus-HR dataset; flowlines with missing slope information or with the minimum slope value (0.00001, imposed by data cutoff) were excluded for the fitting.

However, the resulting $S \sim A$ relationships are highly scattered for most basins. The median coefficient of determination (R^2) is only 0.36, with the 80th percentile reaching 0.52. This indicates substantial uncertainty in the estimated concavity index at the network scale. Consequently, the available data do not allow us to robustly attribute AI- c relationships to concavity-driven changes in slope ratios.

The Spearman correlations among Tokunaga parameter c , basin-averaged slope ratio and area ratio, drainage density and aridity across the 11,946 5th-order Tokunaga self-similar networks analyzed in our main text are shown below, with (***) meaning $p<0.0001$:

	Tokunaga parameter c	AI	Density	Area ratio
AI	0.12 ^{***}			
Density	0.06 ^{***}	0.17 ^{***}		
Area ratio	-0.37 ^{***}	-0.26 ^{***}	-0.32 ^{***}	
Slope ratio	-0.25 ^{***}	-0.44 ^{***}	-0.01	0.50 ^{***}

We have also checked the correlations among Tokunaga parameter c , mean side-branching angle, mean AI, mean area ratio (instead of slope ratio) and mean channel slope for 11,946 5th-order Tokunaga self-similar networks and obtained similar results as using slope ratios in the main text. **Please see Supplementary Table S2.**

Figure 2d: I am puzzled by the slope map. The southern tip of the Appalachians appears to have the similar gradients to the most tectonically active parts of the west coast. This makes me a bit concerned that the size of the hex squares is introducing some bias (i.e., if the hex square is centred on a hilly region, vs a very mountainous region that have half the square on a low relief area (easy to do in, say, southern California), you might get a result not really reflective of the true gradients of the channels. This fear would be mitigated by some analysis of the basin sizes.

Response: We thank the reviewer for raising this critical point. We agree that the mean slope within a fixed-area hexagon can be biased when high-relief terrain occupies only a small fraction of the hexagon. To address this concern, we now use the *median* channel slope rather than the mean slope within each hexagon to display regional slope patterns, thereby substantially reducing sensitivity to outliers and minimizing the influence of mixed low- and high-relief areas. **Please see revised Figure 2d.** We also note that the hexagon map is used solely for visualizing broad spatial patterns; all quantitative analyses in the manuscript are conducted at the individual network scale, not on aggregated hexagon statistics. As a result, any potential biases introduced by hexagon averaging do not affect our main findings.

Additionally, in the revised manuscript, we excluded stream networks in which all the flowlines have a slope of 0.00001 due to the cutoff in the NHDPlus-HR dataset (**see Lines 368-369 of the revised manuscript**), reducing the total number of 5th-order stream networks from 17,021 in the original manuscript to 16,322 in the revised manuscript. The revised analysis yields results that are consistent with those of the original manuscript, demonstrating the robustness of our findings.

Line 186-187: This sentence implies that the 72 degree result is general, and that it has been observed in seepage networks, when in fact the 72 degree result is based specifically on a model including seepage. Do you have a reference for the 72 degrees resulting from a process other than seepage?

Response: We thank the reviewer for this comment. To avoid potential misinterpretation and to improve the overall flow of the text, we have removed the related sentences. **Please see Lines 186-192 in the revised manuscript.**

Line 195: The relationship between slope ratios and delta HS is controlled by the area ratio, so I don't see what the work "also" is doing in this sentence.

Response: We thank the reviewer for pointing out this ambiguity and have revised the sentence to: *"Pairs of tributaries with larger differences in order tend to have larger differences between their upstream slopes (i.e., smaller slope ratios SR) as a direct consequence of the power-law relationship between drainage area and channel slope⁵⁶, combined with Horton's exponential relationship between order and drainage area⁸."* **Please see Lines 222-226 in the revised manuscript.**

Line 284: Add "the" at the beginning of this sentence.

Response: We thank the reviewer for catching this typo and have revised the sentence. We have streamlined our text and revised the sentence to: *"Viewed through this lens, Tokunaga parameter c expresses the prevalence of this side-branching instability: networks in more strongly dissected terrain exhibit higher c values and their junctions exhibit correspondingly stronger contrasts in stream orders, drainage areas (Supplementary Table S2), and channel slopes (Fig. 6)."* **Please see Line 308-312 in the revised manuscript.**

Line 199: Can you explain how the degree of landscape dissection is reflected in the slope ratio? Are you trying to make the point that a more feathered network has greater area ratios at tributaries?

Response: We thank the reviewer for this question. In short, the answer is yes, and the stronger contrasts in drainage area are associated with stronger contrasts in channel slope. We have deleted the sentence at Line 199 of the original manuscript *“These results suggest that the degree of landscape dissection, as reflected by slope ratio, may influence the formation of side branches and consequently shape their topological connectivity and average branching geometry within river networks.”* Indeed, we are emphasizing that more feathered networks exhibit greater contrasts in the drainage areas of upstream tributary pairs. To clarify this point, we revised the section *“Interrelations of stream network topology and planform geometry”* to make the connection clearer: *“Slope ratios, and their corresponding junction angles, are topographic expressions of landscape dissection. A valley's high-order main stream will carry a relatively large discharge and therefore adjust toward a relatively low equilibrium channel gradient³⁴, whereas its side slopes will remain steep unless the streams draining those slopes have sufficient drainage areas (and thus discharges) to incise them⁷. Thus more feathered networks, whose tributary junctions will tend to have larger order differences k and greater contrasts in drainage areas and channel slopes, will also tend to have wider average junction angles. From this perspective, slope ratios and junction angles are not only a direct consequence of local erosion asymmetry, but also a broader result of side-branch formation, which ultimately sets a stream network's topology and geometry.”* **Please see Lines 238-247 in the revised manuscript.**

Line 227: Again, the 72 degrees is the theoretical prediction only for stream sapping.

Response: We agree, and we have removed corresponding sentences.

Line 239-240: “tend to reflect distinct erosion mechanisms”. Reference 53 proposed a very specific erosion model and then went to one of the very few places on Earth where the model could be applied to real topography (Apalachicola is as close to a pile of noncohesive sand as it is possible to get in a natural landscape). And since many of the authors of this paper were also authors of reference 47, they should know that they did not compile erosion mechanisms, but rather speculated on this based on climate regimes. Soften this statement.

Response: We thank the reviewer for this comment. To avoid potential misinterpretation and to improve the overall flow of the text, we have removed the related sentences. **Please see Lines 248-256 in the revised manuscript.**

Line 264: Replace the word “these” with “their” to make it clear this is a result from the Zanardo et al paper.

S Mudd

Response: We thank the reviewer for this suggestion. We opted to retain “these” to preserve a de-personalized formulation and to emphasize the results, as the reference to Zanardo et al. is explicit

in this sentence. To make it clear, we revised the sentence to include the citation directly: “*However, these relationships weakened or disappeared when coarser channel initiation thresholds (i.e., drainage area $\geq 0.3 \text{ km}^2$) were used²⁴, raising the question of whether climatic signatures persist in real-world network topology.*” Please see Lines 277-280 in the revised manuscript.

REVIEWERS' COMMENTS

Reviewer #1 (Remarks to the Author):

8 February 2026

Second Review of “Climate’s influence on topography encoded in stream network topology and geometry”

I have now read through the revised version of this manuscript, including the supplement and response to reviewer comments. I believe the revised manuscript represents an improvement upon the original submission, and I am satisfied with the author responses to my previous comments (and corresponding changes to the text). Overall, I think this is an interesting contribution that can be published as is. At this point, I only have a few very minor comments that I have listed below, and which the authors can choose to take or leave as they see fit. I’d like to thank the authors for their thorough response to my previous comments, and for sharing their exciting science.

Response: We appreciate the reviewer’s positive assessment of our work. We have carefully considered the minor comments provided and addressed them where appropriate, as detailed in the responses.

Minor comments:

L214-223: Parts of this explanation were a bit hard for me to follow and I had to take my time with it. Here’s how I think about it (which I think may be a bit more intuitive than the text in the manuscript). Side-branches need to have higher contrasts in slopes (lower SR) than bifurcations due to the scaling of slope with drainage area. This dictates that side-branching angles should generally be wider than bifurcation angles, since a wider angle minimizes the portion of the main valley along which the side branch flows. Consider a thought experiment where a side branch comes in with a tiny (let’s say 1 degree) junction angle with the main stem. In this case, both the side branch and the main stem must be flowing along the central valley axis. Thus, the topography would need to be really strange for the side branch to have a steeper slope than the main stem. But, if the side branch doesn’t have a steep slope, there’s likely not enough discharge (for that low drainage area of the side branch) to move the imposed sediment load. In contrast, if the side branch comes in at a 90-degree junction angle, there is no overlap between the side branch and the main valley axis, allowing a low SR. Aspects of this come up in L238-247, but even then, I don’t feel like those lines fully explain this concept.

Response: We understand the reviewer's thought experiment, but we are cautious about basing the discussion on it, since its application to low-gradient landscapes seems unclear to us. We did add, **at line 186 of the revised manuscript**, the point that large differences in stream order imply large

contrasts in both drainage area and slope. This was already mentioned at line 227 of previous version (now at line 196 of the revised manuscript) but now we mention it both places.

Fig. 4: In Fig. 4, red colors correspond to high AI, which is a humid climate. I (and I think most other people?) tend to think of red as warm/arid. It might help to flip the color scheme for 4a and 4c and make red the low AI (arid) color and make the high AI (humid) bin a cooler color (blue?).

Response: We thank the reviewer for this helpful suggestion. We have revised Fig. 4 to use dark blue for more humid conditions and light blue for more arid conditions, improving accessibility for color-blind readers. Notably, different symbols are also used to distinguish aridity classes.

While I found the authors analyses to be robust, one thing that struck me, especially with Figure 7, is that drainage density isn't discussed in the manuscript. Doesn't the increase in Tokunaga parameter c with AI (higher c in more humid climate) imply an increase in drainage density (i.e., higher drainage density in more humid climate)? If so, this seems at odds with studies showing drainage density decreases with increases in mean annual precipitation (e.g., Chadwick et al, 2013). This could be worth a few lines of discussion in the manuscript.

Chadwick, Oliver A., Josh J. Roering, Arjun M. Heimsath, Shaun R. Levick, Gregory P. Asner, and Lesego Khomo. 2013. "Shaping Post-Orogenic Landscapes by Climate and Chemical Weathering." *Geology* 41 (11): 1171–74.

Response: A larger value of Tokunaga parameter c means that low-order streams are more likely to join higher-order streams (big order differences k) than they are to join other low-order streams (small order differences k). A larger value of Tokunaga parameter c does not mean that there are more low-order streams overall; that is set by the bifurcation ratio instead. Nor does Tokunaga parameter c determine the relative lengths of lower order streams, which instead is set by the length ratio. Thus changes in drainage density will be determined by the joint effects of the bifurcation ratio and the length ratio, not by Tokunaga parameter c . Note, this does not mean that there cannot be an indirect correlation between topological and geometric characteristics of channel networks.

Reviewer #3 (Remarks to the Author):

Thanks to the authors for their considered and substantive revisions to this submission. I have no further comments, and am happy to see this work advance.

Response: We thank the reviewer for their careful review and positive assessment of our revised manuscript.

Reviewer #4 (Remarks to the Author):

I am satisfied with the authors' responses to my previous comments. However they have added some text in response to another reviewer that I don't agree with (and mis-represent the findings from one of my papers). However this is easily corrected and constitute minor revisions.

Response: We thank the reviewer for their careful reading and for their positive assessment of our previous responses. We apologize for misrepresenting the findings from the reviewer's work. We have revised the relevant text.

Line 62: Insert "The" before Tokunaga. Same on line 126.

Response: One of our native English speakers (James Kirchner) holds that the definite article "the" here would be misleading and confusing because there is no other "parameter c " under discussion. Using the definite article, as in "the Tokunaga parameter c ", will lead readers to ask themselves, "as compared to which other parameter c ?", for which they will not find an answer because there is none. This can of course be revisited if the copyeditor believes a definite article is needed here. **Please note that line 62 and line 126 in the previous version now correspond to lines 62 and 116 in the revised manuscript.**

Line 133-134: Say which channels are being compared the SR here (not just in methods).

Response: We thank the reviewer for this helpful suggestion. The original sentence at Lines 133-134 said "*Here, slope ratio is defined as the ratio of the gentler to the steeper slope and therefore ranges from 0 to 1 (see Methods).*" We have now revised it to "*Here, slope ratio is defined as the ratio of the gentler to the steeper slope in each pair of upstream tributaries and therefore ranges from 0 to 1 (see Methods).*" **Please see Lines 123-125 in the revised manuscript.**

Lines 159-162: Consider making a comparison to reference 24 here.

Response: We did not add a comparison to ref. 24 in Lines 159–162 (**now Lines 150-153 in the revised manuscript**); instead, the manuscript already addressed this comparison in detail later in the "*Drivers of stream topology and geometry*" section, where it fits more naturally with the analysis. **Please see Lines 227-248 in the revised manuscript:** "*Prior studies have hinted at possible climatic influences on stream network topology. For example, ref. 24 reported correlations between Tokunaga parameter c and precipitation or storm frequency across 408 stream networks extracted from digital elevation models in the United States. However, these relationships weakened or disappeared when coarser channel initiation thresholds (i.e., drainage area ≥ 0.3 km²) were used²⁴, raising the question of whether climatic signatures persist in real-world network topology. Our partial correlation analysis (Fig. 6) indicates that climatic aridity has only a weak direct influence on parameter c ($\rho_{\text{partial}}=-0.06$), but stronger relationships with average channel slope ($\rho_{\text{partial}}=0.28$), slope ratio ($\rho_{\text{partial}}=-0.32$), and mean side-branching angle*

($\rho_{\text{partial}}=0.22$). These variables, in turn, are correlated with parameter c . This is consistent with the concept that climatic aridity affects channel incision, which in turn controls landscape dissection⁷, thereby setting Tokunaga parameter c , channel slopes and slope ratios, which ultimately impact side-branching angles. After accounting for climate aridity, slope ratio, and mean side-branching angle, mean channel slope becomes a stronger predictor of Tokunaga parameter c (increasing from an overall correlation of $\rho=0.06$ to a partial correlation of $\rho_{\text{partial}}=0.19$). This suggests that channel slope exerts a more substantial direct influence on parameter c than climatic aridity does (and that this influence is masked in the correlations reported in ref. 24 and shown in Fig. 2g). These relationships suggest that the observed Spearman correlation between climatic aridity and Tokunaga parameter c (Fig. 2e) is primarily mediated through climate effects on topography and network geometry (Fig. 6) rather than through a direct impact of climate on network topology.”

Line 220-231: Reference 50 has a long discussion of these effects so I am surprised that paper is not cited here.

Response: We agree and have added the citation to the respective sentence. Please note that lines 220–231 in the previous version now correspond to lines 187–199 in the revised manuscript; the specific addition can be found **at line 189 of the revised manuscript**.

Line 252: Reference 50 showed that the junction angles were a function of the area ratio, which could have a large range in “side-branching” channels. Lumping all side branching channels seems like a good way to hide interesting behaviour.

Response: Figure 5 and the associated discussion (**lines 249–255 in the previous version, now lines 210–218 in the revised manuscript**) refer to distributions of branching angles and how those distributions change with Tokunaga parameter c . By definition, distributions involve lumping channels together (otherwise there is no distribution). Of course there will also be reasons for the variations that one will see if one looks at individual branching angles within these distributions, but that is not the point of Figure 5. Figure 3 and associated discussion (**Lines 182-199 in the revised manuscript**) provide more details on side branches with varying Horton-Strahler order differences. We have modified **line 189 in the revised manuscript** to say, "*This directly implies greater contrasts in channel slopes at side-branching junctions ($k>0$) than at bifurcation junctions ($k=0$), and greater contrasts in drainage area (and thus channel slope⁵⁰) contrasts in networks with greater degrees of side-branching (as measured by higher values of Tokunaga parameter c).*"

Line 254-256: I don't agree with these statements, and it mis-cites our paper. Lateral erosion does not produce low-order side branches. In the case of very large area ratios, where the larger channel meanders, there is a greater probability that a side tributary will intersect with a meander apex. Here is the text from our paper (reference 50):

The complex hydrodynamics of river confluences and associated sedimentation and erosion (e.g., ref. (69) and references therein) may plausibly promote the lateral migration of network junctions in the direction of the smaller tributary, causing the larger river to form a bend, with the tributary joining at the outside apex. This explanation seems unlikely for two reasons. First, our junction angle measurements reflect the network-scale structure rather than the functional scale of confluence hydrodynamics. Second, it seems inconceivable that the paths of very large channels could be meaningfully impacted by channels orders of magnitude smaller.

What appears as the “deflection” of large rivers by minor tributaries may alternatively be interpreted as minor tributaries preferentially joining large rivers at the outside apex of large bends. In other words, the location of the junction may be influenced by the location of the bend, rather than vice-versa. This interpretation is supported by the observation that developing tributaries nucleate at the outside apex of bends when tidal channel networks form (70).

Response: We thank the reviewer for pointing this out. We apologize for misrepresenting reference 50 in Lines 254-256 (now Lines 216-218 in the revised manuscript) and we have removed the citation accordingly. Please see Lines 216-218 in the revised manuscript.